# 1 Temperature-RH Dependent Viscosity of Organic Aerosols

# 2 from 273 to 303 K: Implications for Global N<sub>2</sub>O<sub>5</sub> Uptake

- 3 Atta Ullah<sup>1</sup>, Ying Li<sup>2</sup>, Mijung Song<sup>1,3</sup>
- <sup>1</sup>Department of Earth and Environmental Sciences, and Earth Environmental System Research Center, Jeonbuk
- National University, Jeollabuk-do Jeonju-si 54896, Republic of Korea; <sup>2</sup> Key Laboratory of Industrial Ecology
- and Environmental Engineering (Ministry of Education), School of Environmental Science and Technology,
- Dalian University of Technology, Dalian 116024, China; <sup>3</sup>Department of Environment and Energy, Jeonbuk
- National University, Jeollabuk-do Jeonju-si 54896, Republic of Korea
- Correspondence to: Mijung Song (Mijung.Song@jbnu.ac.kr)

# Abstract

Organic aerosol (OA) viscosity and phase state govern multiphase diffusion and reactivity, yet systematic constraints across tropospheric temperature (T)–relative humidity (RH) space remain limited. We measured the viscosity of sucrose–H<sub>2</sub>O droplets (OA surrogate) over 273–303 K and ~20–90% RH using bead-mobility and poke-and-flow methods, spanned ~9 orders of magnitude. A Vogel–Fulcher–Tammann fit with experimentally derived fragility ( $D_f$  = 13) extended the parameterization to 230–310 K and 0–100% RH. When coupled with zonal-mean tropospheric T–RH fields (2020–2024), the parameterization yielded global distributions of viscosity and organic-phase mixing time ( $\tau_{mix,org}$ ) for 200-nm particles: liquid states prevailed below ~2 km, semisolid regimes occupied ~2–9 km (latitude dependent), and near-glassy conditions emerged above ~9 km;  $\tau_{mix,org}$  was <1 h in the boundary layer but frequently exceeded 1 h aloft. Calculations indicated the N<sub>2</sub>O<sub>5</sub> uptake coefficient was generally  $\geq$ 10<sup>-2</sup> for liquid particles in the boundary layer, decreased by ~1–2 orders above ~2–4 km as bulk diffusion became rate-limiting; with surface hydrolysis, N<sub>2</sub>O<sub>5</sub> uptake coefficient leveled near ~10<sup>-3.5</sup> aloft, and without it can drop to 10<sup>-5</sup>–10<sup>-6</sup> at viscosity  $\gtrsim$  10<sup>9</sup> Pa·s. These results highlight the need for temperature-sensitive viscosity in next-generation air-quality and climate models.

1 Introduction

Organic aerosols (OAs), including both primary (POA) and secondary organic aerosol (SOA), are ubiquitous in the Earth's atmosphere. These aerosols play vital roles in climate, air quality, and human health through their interactions with solar radiation, cloud microphysics, and heterogeneous atmospheric chemistry (Pandis et al., 1992; Pöschl, 2005). In the troposphere, OA is frequently exposed to a wide range of relative humidity (RH) and temperature, conditions that drive phase transitions that profoundly modify its physicochemical properties. Depending on chemical composition, RH, and temperature, OA can exist in liquid, semi-solid, or solid (or glassy) states (Koop et al., 2011; Reid et al., 2018). Dynamic viscosity serves as a useful parameter for classifying these states: viscosity  $< 10^2$  Pa·s corresponds to a liquid phase,  $10^2 \le viscosity \le 10^{12}$  Pa·s to a semi-solid phase, and viscosity > 10<sup>12</sup> Pa·s a glassy phase (Koop et al., 2011; Reid et al., 2018). Accurate determination of aerosol viscosity and phase state is critical for predicting heterogeneous reaction kinetics (Zhou et al., 2013; Marshall et al., 2018), particle size distributions (Shiraiwa and Seinfeld, 2012; Shiraiwa et al., 2013; Zaveri et al., 2018; Song et al., 2022), and ice nucleation (Knopf et al., 2018; Ladino et al., 2014; Lienhard et al., 2015). Highly viscous SOA matrices impede molecular diffusion, suppress oxidant uptake, and finally can alter SOA yields (Abbatt et al., 2012; Gržinić et al., 2015a; Kuwata and Martin, 2012). Over the past two decades, laboratory and modeling studies have established RH as a primary control on OA viscosity at near-room temperature, transitioning from liquid to semi-solid or glassy as RH decreases (Song et al., 2015a; Song et al., 2016a; Song et al., 2019a; Song et al., 2016b; Song et al., 2021b; Gerrebos et al., 2024; Gregson et al., 2023; Kiland et al., 2023; Nikkho et al., 2024; Grayson et al., 2016; Grayson et al., 2017; Saukko et al., 2012). The role of chemical compositions on OA viscosities has also been widely explored, showing that parameters such as oxygen-to-carbon ratio, molar mass, functional group and volatility diversity can significantly affect the phase state and viscosity of OA (Derieux et al., 2018; Grayson et al., 2017; Rothfuss and Petters, 2017a; Li et al., 2020; Champion et al., 2019; Shiraiwa et al., 2017). However, despite substantial progress on RH- and

composition-dependence, the explicit role of temperature across the tropospheric range (230 - 310 K) remains poorly constrained. While there have been modeling efforts to predict viscosity changes with temperature using parameterizations such as the fragility parameter or Vogel-Fulcher-Tammann (VFT) frameworks (Kiland et al., 2023; Rothfuss and Petters, 2017b; Gerrebos et al., 2024; Kasparoglu et al., 2021), experimental studies that directly probe temperature-dependent viscosity remain scarce. This gap in temperature-dependent viscosity data has significant implications for heterogeneous chemistry. For instance, high viscosity can suppress reactive gas uptake by several orders of magnitude (Shiraiwa et al., 2011; Marshall et al., 2016). Modeling and field observations indicated that viscous organic shells can reduce the N<sub>2</sub>O<sub>5</sub> uptake coefficient by 1-3 orders of magnitude (Song et al., 2025a), while liquefaction can restore reactivity by a factor of 10 to 100 (Gržinić et al., 2015a; Hallquist et al., 2003). Therefore, improving our understanding of OA viscosity depending on atmospheric condition including temperature variations is critical for accurately representing heterogeneous processes in atmospheric models. To address this knowledge gap, we conducted a systematic experimental investigation of OA viscosity under tropospherically relevant temperature and RH conditions, utilizing sucrose as a well-characterized OA model compound (Power et al., 2013; Zobrist et al., 2011; Bateman et al., 2014; Kiland et al., 2019). Viscosity was measured at 273, 283, 293, and 303 K and 20 - 90% RH using bead-mobility and poke-and-flow techniques. The measured viscosity data were parameterized to construct a T-RH phase diagram for sucrose-H<sub>2</sub>O droplets. Based on parameterized viscosity, we analyzed global zonal-mean profiles of viscosity and the corresponding mixing times of organic molecules ( $\tau_{mix,org}$ ) within 200 nm sucrose particles. Finally, we predicted heterogeneous N<sub>2</sub>O<sub>5</sub> uptake coefficient across tropospheric T-RH conditions using the viscosity fields in a mechanistic framework that accounts for bulk diffusion and surface hydrolysis.

2 Experimental and Methods 74 2.1 Materials and aerosol generation 75 Sucrose (99.5% purity, Sigma-Aldrich) was dissolved in high-purity water (18 M $\Omega$ ·cm, Merck Millipore Synergy, 76 Germany) to prepare approximately 10 wt.% aqueous solutions. In all experiments, sucrose droplets were 77 generated by nebulizing an aqueous solution with a low-internal-volume glass nebulizer (Meinhard, USA) and 78 depositing them onto a hydrophobic substrate (Hampton Research, Canada). Using the droplets, we systematically 79 determined their viscosity across four temperatures (273, 283, 293, and 303 K) and a wide range of RH conditions 80 (20-90%) via bead-mobility, and poke-and-flow. 81 2.2 Viscosity measurements by bead-mobility technique ( $\eta \approx 10^{-3}$  -  $10^{3}$  Pa·s) 82 The bead-mobility technique has been described in detail elsewhere (Renbaum-Wolff et al., 2013a; Jeong et al., 83 2022; Song et al., 2015a; Renbaum-Wolff et al., 2013c); here, we briefly summarize the procedure as applied in 84 this work, Insoluble melamine beads (~1 um, Cat. No. 86296, Sigma-Aldrich) were nebulized into sucrose-H<sub>2</sub>O 85 droplets deposited on a hydrophobic substrate (diameter: ~40–100 μm) in a flow-cell (TSA12Gi, INSTEC, USA). 86 A total N<sub>2</sub>/H<sub>2</sub>O gas flow of 1.2 L min<sup>-1</sup> generated a uniform shear stress on the droplet surfaces at high RH (60 – 87 90%) and temperature (273 - 303 K). 88 Prior to each measurement, the droplets were equilibrated at the target RH and temperature for sufficient time 89 to ensure thermodynamic conditioning. At each temperature, RH was systematically reduced every ~5 % from 90 ~90 to ~60 %, with the exact RH steps varying between experiments. During the experiment, the bead motion in 91 a droplet was recorded with an optical microscope (Olympus CKX53 with 40 × objective, Japan) and CCD camera 92 (Hamamatsu C11440-42U30, Japan). The bead velocities were then converted to viscosity using a calibration 93 curve (Fig. S1). Figure S2 shows representative bead movements within sucrose droplets at different temperatures. 94 The corresponding mean bead speeds as a function of temperature and RH are presented in Fig. S3a, and the

al., 2021a).

derived viscosities obtained using the calibration curve (Fig. S1) are shown in Fig. S3b. When the bead movements 96 were too slow to measure, the poke-and-flow technique was applied. 97 2.3 Viscosity measurements by poke-and-flow ( $\eta > \sim 10^3 \text{ Pa} \cdot \text{s}$ ) 98 The poke-and-flow technique is widely used to determine the viscosity of highly viscous single droplets (Grayson 99 et al., 2015; Gerrebos et al., 2024; Song et al., 2015b; Renbaum-Wolff et al., 2013c). In this study, sucrose-H<sub>2</sub>O 100 droplets with diameters ranging from ~20 to ~40 μm on a hydrophobic glass substrate were conditioned in the 101 flow-cell. At each temperature (273, 283, 293, and 303 K), RH was systematically reduced from approximately 102 60% to lower levels in discrete steps (typically ~5%), and the poke-and-flow method was applied until either 103 viscous flow or particle fracture was observed. The required conditioning time of each droplet for thermodynamic 104 equilibrium was determined for each RH-T setting and are described in Section S1 and Table S1. After 105 equilibration, each droplet was poked using a fine needle (Jung Rim Medical Industrial, South Korea), mounted 106 on a micromanipulator (Narishige, model MO-152, Japan). The morphological evolution of the droplets before, 107 during, and after poking was monitored via optical microscopy (Olympus CKX53 with a 40× objective, Japan) 108 and captured using a CCD camera (Hamamatsu, C11440-42U30, Japan). 109 The experimental flow time,  $\tau_{exn.flow}$ , was defined as the time required for the diameter of the central cavity to 110 reduce to 50% of its value immediately after poking. Figure S4 presents representative optical images showing 111 the morphological evolution of a sucrose-H<sub>2</sub>O droplet at ~50% RH for 273 - 303 K during the experiment. Averaged  $\tau_{exp flow}$ , reflecting temperature and RH dependence are summarized in Fig. S5a. These values of sucrose-112 113 H<sub>2</sub>O droplets were then converted to the viscosity using the equation proposed by Sellier et al. (2015) resulting 114 viscosities shown in Fig. S5b. At lower RH, brittle cracking without relaxation after poking was observed as 115 shown in Fig. S6. When restorative flow was not detected after more than 2 h, we assigned the viscosity as  $\geq$ 116  $1 \times 10^8$  Pa·s (Song et al., 2021b; Jeong et al., 2022; Song et al., 2019b; Renbaum-Wolff et al., 2013c; Maclean et 117

### 2.4 Determination of fragility parameter and glassy phase transition

To determine the fragility parameter ( $D_f$ ), which describes how sensitively viscosity responds to temperature near the glass transition, we fitted our experimental viscosity data—spanning a temperature range of 273 – 303 K and RH ~20 – 90% — to the VFT equation Eq. (1).

$$ln \, \eta(\text{RH, T}) = ln \, \eta_{\infty} + \frac{D_f T_o(RH)}{T - T_o(RH)} \tag{1}$$

where  $\eta(RH, T)$  is the viscosity at a given RH and temperature,  $\eta_{\infty} = 1 \times 10^{-5} \, \text{Pa·s}$  is the viscosity at infinite temperature, (Angell, 1991) and  $T_0(RH)$  is the RH-dependent Vogel temperature, which can be obtained by rearranging Eq. (1):

$$T_{o}(RH) = \frac{\ln\left(\frac{\eta(RH, 293K)}{\eta_{\infty}}\right)(293K)}{D_{f} + \ln\left(\frac{\eta(RH, 293K)}{\eta_{\infty}}\right)}$$
(2)

where  $\eta(RH, 293 \text{ K})$ , is the viscosity at 293 K, estimated using a mole-fraction-based Arrhenius mixing rule. Fitting this relation yielded a hygroscopicity parameter  $k = 0.061 \pm 0.0023$  (Fig. S7), and full parametrization details are provided in Section S2. Figure S8 shows the resulting best-fit value for  $D_f$  as  $13 \pm 1$ , which is derived directly from temperature-dependent viscosity measurements. While prior studies often relied on assumed or literature-based  $D_f$  values (generally ~10) for various SOA types (Kiland et al., 2023; Maclean et al., 2021b; Gregson et al., 2023; Derieux et al., 2018), this work provides the first experimentally constrained estimate of  $D_f$  based on the temperature dependent viscosity data for sucrose-H<sub>2</sub>O droplets.

#### 2.5 Simulation of N2O5 uptake coefficient

Simulations of the  $N_2O_5$  uptake coefficient are based on the resistor model,(Ammann et al., 2013; Gržinić et al., 2015b) which has been detailed in our previous study (Song et al., 2025b). Principally,  $N_2O_5$  uptake coefficient is calculated considering surface accommodation, surface reaction, transfer from surface to bulk, and the bulk reaction-diffusion resistance. As shown in Eq. (3), the surface accommodation coefficient ( $\alpha_s$ ) is set to be 1 assuming that surface accommodation is not rate limiting.  $\Gamma_s$  represents the limiting uptake coefficient for the surface reaction, which was estimated to be  $2.5 \times 10^{-4}$  based on previous simulations and observations at low RH conditions (Gržinić et al., 2015b).  $1/\Gamma_{sb}$  represents the resistance for surface to bulk transfer, calculated by  $1/\Gamma_{sb} = 1/\alpha_b - 1/\alpha_s$ , where  $\alpha_b$  is set to be 0.035 (Gržinić et al., 2015b; Song et al., 2025b).  $1/\Gamma_b$  represents the bulk reaction-diffusion resistance given by Eq. (4).

$$\frac{1}{\gamma_{N_2O_5}} = \frac{1}{\alpha_s} + \frac{1}{\Gamma_s + \left(\frac{1}{\Gamma_{sb}} + \frac{1}{\Gamma_b}\right)^{-1}} \tag{3}$$

$$\frac{1}{\Gamma_b} = \frac{\omega}{4HRT\sqrt{D_{N205}k^I}} \left(\coth q \cdot \frac{1}{q}\right)^{-1} \tag{4}$$

Values of the mean thermal velocity of  $N_2O_5$  ( $\omega$ ), the Henry's law constant (H), and the hydrolysis rate ( $k^{\rm I}$ ) in Eq. (4) are taken from our previous study (Song et al., 2025b). R is the gas constant. The reacto-diffusive parameter (q) is calculated as r/l, where r is the radius of sucrose particles, assumed to be 100 nm in this study. l is the reacto-diffusive length, defined by  $\sqrt{D_{\rm N2O5}/k^{\rm I}}$ , where  $D_{\rm N2O5}$  is the diffusivity coefficient of  $N_2O_5$  in OA calculated from the fractional Stokes-Einstein equation as described in Section S1.

3 Results and discussion 151 3.1 Temperature and RH dependent-viscosities of sucrose-H<sub>2</sub>O droplets 152 Figure 1 illustrates the effect of temperature on the viscosity of sucrose-H<sub>2</sub>O droplets across a range of RH. In this 153 study, we systematically investigated viscosity changes at four specific temperatures—273, 283, 293, and 303 154 K—across a 30 K range and RH values from ~20 to ~90%. The result shows that viscosity of the droplets varied 155 by approximately nine orders of magnitude, suggesting the strong temperature sensitivity modulated by RH. At 156 293 K, the current result agrees with previous studies (Jeong et al., 2022; Song et al., 2021a; Song et al., 2016a; 157 Song et al., 2015a; Power et al., 2013; Grayson et al., 2015; Renbaum-Wolff et al., 2013b), confirming the 158 robustness of our measurements. At other temperature, however, viscosity data are scare, and thus our study 159 provides the first systematic benchmarks across 273 – 303 K, filling a critical gap in the literature. 160 As temperature increased from 273 to 303 K, viscosity consistently decreased, with the magnitude of reduction 161 depending on RH. For example, at high RH ( $\sim$ 74 – 80 %), viscosity decreased by approximately one order of 162 magnitude—from ~1.2×10<sup>1</sup> Pa·s at 273 K to ~1.9×10<sup>0</sup> Pa·s at 303 K, which correspond to a liquid state (Fig. 1). 163 At mid RH (~50%), the temperature increase caused a much larger reduction, from ~2.1 × 10<sup>6</sup> Pa·s at 273 K to 164 ~1.4 × 10<sup>3</sup> Pa·s at 303 K, indicating a semi-solid state. This behavior was directly observed in optical images 165 during poke-and-flow experiments (Fig. S4), which revealed distinct flow-time contrasts between the temperature 166 ranges. 167 At lower RH, the droplets exhibited brittle cracking by poking at all temperatures. The cracking RH threshold 168 decreased systematically with increasing temperature: ~38% RH at 273 K, ~33% at 283 K, ~24% at 293 K, and 169 ~18% at 303 K (Fig. S6). Once fracture occurred, no restorative flow was observed for more than two hours, 170 indicating a near- glassy state (viscosity  $> 1 \times 10^8 \, \text{Pa} \cdot \text{s}$ ) (Song et al., 2021b; Jeong et al., 2022). The viscosity 171 difference associated with a 30 K temperature shift becomes increasingly pronounced at lower RH, highlighting 172

the combined sensitivity of OA phase behavior to both temperature and RH.

Song et al., 2021a; Jeong et al., 2022).

Figure 1: Viscosity data from bead mobility and poke-and-flow experiments at  $273 \pm 1$  K (magenta circles),  $283 \pm 1$  K (blue circles),  $293 \pm 1$  K (red circles) and  $303 \pm 1$  K (green circles) across a RH range of 20 - 90%. The x-axis error bars represent the uncertainty in RH from the RH sensor during calibration at each temperature, while the y-error bars show the standard deviation in the measured viscosity calculated from 3 - 5 beads across 2 - 5 particles. Dotted lines show linear fit of viscosity versus RH at each temperature, with line colors matching the symbol colors for the corresponding temperatures. Experimental data from previous studies are included for comparison (Power et al., 2013; Rothfuss and Petters, 2017b; Renbaum-Wolff et al., 2013b; Grayson et al., 2015; Song et al., 2015a; Song et al., 2016a;

We then extended the results with the VFT equation to calculate viscosity across a wider range of RH (0 – 100 %) and temperature (230 – 310 K), using the fragility parameter ( $D_f$  = 13) directly derived from our experimental measurements. Figure 2 illustrates the resulting RH–temperature phase diagram, in which contour lines (isopleths) of constant viscosity.

The orientation of these isopleths varies systematically with temperature. At low temperatures (T 

Figure 2: Phase diagram showing viscosity isopleths for sucrose- $H_2O$  droplets as a function of temperature and RH. Viscosity isopleths were computed using the VFT equation, with the  $D_f$  obtained from experimental data to VFT equation (Fig. S8). Overlaid circles show the experimental data points for viscosity measured at specific temperature and RH conditions as shown in Fig. 1. The black dash line indicates the transition from liquid to semi-solid of sucrose- $H_2O$  droplet. The brown area represents the temperature and RH conditions where the sucrose- $H_2O$  droplets exhibit a glassy state.

#### 3.2 Viscosity and mixing times of sucrose-H<sub>2</sub>O droplets at a range of tropospheric conditions

Building on the phase diagram in Fig. 2, we applied the viscosity parameterization to reanalysis-based RH and temperature fields to evaluate realistic tropospheric conditions. Figure 3 presents the zonal-mean distributions of (a) viscosity and (b)  $\tau_{mix,org}$  within 200 nm of sucrose droplets as a function of latitude and altitude, derived from monthly mean ambient RH and temperature from 2020 to 2024 (Fig. S9 and Section S3).  $\tau_{mix,org}$  was calculated from the viscosity values using the Stokes–Einstein equation as described in the Section S1 (Eq. S1.5),

representing the time required for organic molecules to equilibrate within a 200 nm droplet and thus indicating the time of internal mixing. In Fig. 3a, zonal-mean viscosity values of sucrose-H<sub>2</sub>O droplets revealed a clear stratification: liquid phase states dominate below ~2 km, semi-solid states exist between ~2 and ~9 km depending on latitude, and glassy states emerge above  $\sim 9$  km. At high latitudes (> 70 °) in southern hemispheres, liquid phases extended to  $\sim 3-4$ km above the surface (Fig. 3a), higher than in the tropics and northern hemisphere polar region where liquid states are typically confined below ~2 km. This persistence is consistent with the extremely high RH observed in these regions (Fig. S9), which enhances water uptake and plasticization of the organic matrix, thereby suppressing viscosity. Semi-solid regimes also extended to higher altitudes in the polar regions than in the midlatitudes, reflecting the moderating influence of RH against the transition to glassy states. Compared with Shiraiwa et al. (2017) who predicted SOA phase state with a global chemistry climate model (EMAC-ORACLE) using regionally and vertically resolved meteorology, our estimates are derived from zonal-mean tropospheric T-RH fields and a single-component sucrose-H2O surrogate. Zonal averaging damps low-RH/low-T extremes that promote glassy states, and sucrose is strongly plasticized by water; together these factors tend to diagnose lower Tg/T (i.e., less viscous) than the multi-component SOA fields reported. Using the viscosity data of sucrose-H<sub>2</sub>O droplets and the Stokes-Einstein equation, Fig. 3b shows the calculated  $\tau_{mix,org}$  of organic molecules in 200 nm sucrose droplets. Results indicate that the  $\tau_{mix,org}$  of sucrose droplets was shorter than 1 hour at all latitudes in the lower troposphere ( $

233

234

235

236

237

Our analysis focuses on sucrose as a proxy system, but real atmospheric aerosols are chemically more complex and often internally mixed with inorganic salts or primary organics. Inorganic components, owing to their hygroscopicity, would generally lower viscosity by enhancing water uptake, thereby shortening  $\tau_{mix,org}$  under humid conditions. Further work is therefore required to investigate viscosity and mixing times in multi-component aerosol systems.

Figure 3: Annual zonal-mean global profiles for (a) viscosity (b) mixing times of organic molecules, and (c) N<sub>2</sub>O<sub>5</sub> uptake coefficient in 200 nm sucrose droplets as a function of altitude and latitude based on average annual RH and temperature fields for the years 2020 to 2024, obtained from Copernicus Climate Data Store (https://cds.climate.copernicus.eu/). The purple dashed line in panel (a) shows the viscosity for 10<sup>2</sup> Pa·s. The green dashed line in (b) shows the mixing time of 1 hr. Seasonal variability of the viscosity and mixing times for sucrose-H<sub>2</sub>O droplets is shown in Fig. S10.

### 4 Atmospheric implications

#### 4.1 Global N2O5 uptake

Viscosity-induced diffusion limitation has direct consequences for heterogeneous chemistry. For example, highly viscous sucrose matrices suppress the transport of  $N_2O_5$  and water into particle bulk, potentially lowering reactive uptake coefficients by orders of magnitude. Conversely, under warm and humid conditions, liquefaction increases diffusivity, enhancing uptake. Using our parameterized viscosity fields, we estimated that in the midlatitude upper troposphere, suppressed uptake could extend over several shorts of the year, particularly in winter. This aligns with recent field observations showing reduced  $N_2O_5$  loss rates in cold conditions (Zhang et al., 2025; Wagner et al., 2013). While these results are presented here as a case study, they illustrate the strong temperature dependence of multiphase reactivity.

For liquid particles, the  $N_2O_5$  uptake coefficient of is generally higher than  $\sim 10^{-2}$ , and is most prevalent in the atmospheric boundary layer. When altitude exceeds  $\sim 2^{-4}$  km depending on latitudes,  $N_2O_5$  uptake coefficient decreases by 1-2 orders of magnitude, implying that the  $N_2O_5$  uptake rates of can be limited by slow bulk diffusion within viscous particles, which may lead to decreased concentrations of particulate nitrates or increased gas-phase  $NO_3$  (You et al., 2012). On highly viscous or glassy particles, surface hydrolysis can dominate the uptake. As shown in Fig. 3c,  $N_2O_5$  uptake coefficient is leveled on  $\sim 10^{-3.5}$  when altitudes continue to increase, as setting the value of the surface reaction term ( $\Gamma_8$ ) is  $2.5 \times 10^{-4}$  in Eq. 3. A sensitivity simulation without considering

the surface hydrolysis ( $\Gamma_s$  is set to be 0 in Eq. 3),  $N_2O_5$  uptake coefficient can be decreased to as low as  $10^{-5}$  –  $10^{-6}$  when the viscosity is higher than  $\sim 10^9$  Pa s (Fig. S11), in which case  $N_2O_5$  uptake is only limited by bulk diffusivity, and hence particle viscosity (Gržinić et al., 2015b; Song et al., 2025b).

By directly constraining temperature-dependent viscosity across a 30 K range and coupling it to global RH–temperature climatology, this study provides one of the first systematic experimental datasets for assessing OA phase behavior under realistic tropospheric conditions. These results highlight the necessity of incorporating temperature-sensitive viscosity parameterizations into next-generation air quality and climate models to more accurately predict particle lifetimes, reactivity, and cloud interactions.

**Author Contributions** 

| 272 | M.S. designed this study. A.U. and M.S. conducted viscosity experiments and analyzed the data. Y.L. calculated |
|-----|----------------------------------------------------------------------------------------------------------------|
| 273 | the reactive uptake of N <sub>2</sub> O <sub>5</sub> . A.U., Y.L., and M.S. wrote the manuscript.              |
| 274 | Funding Sources                                                                                                |
| 275 | This work was supported by the National Research Foundation of Korea (NRF) grant funded by the Korea           |
| 276 | government (MSIT) (RS-2024-00335536), by Global - Learning & Academic research institution for Master's        |
| 277 | PhD students, and Postdocs (LAMP) Program of the NRF grant funded by the Ministry of Education (No. RS-        |
| 278 | 2024-00443714), and by National Natural Science Foundation of China for funding (Nos. 42330605 and             |
| 279 | 42475124).                                                                                                     |
| 280 | Notes                                                                                                          |
| 281 | The authors declare no competing financial interest.                                                           |
| 282 | Acknowledgment                                                                                                 |
| 283 | We thank Gyoung Hee Go for technical support.                                                                  |
| 284 |                                                                                                                |

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
