# Peer review of "Temperature-RH Dependent Viscosity of Organic Aerosols"

_EGUsphere, 2025_

## Referee Comment (RC2)

**Overall:**

The manuscript by Ullah et al. reports new measurements of aerosol dynamic viscosity. Viscosity was determined for particles composed of sucrose—water, a commonly used proxy for atmospheric organic aerosols, at discrete relative humidity (RH) and temperature (T) conditions using established techniques. Notably, the measurements of sucrose particle viscosity as a function of temperature are novel and represent a valuable contribution to the literature. The authors apply their experimental data within the Vogel—Fulcher—Tammann (VFT) framework and the Arrhenius mixing rule to parameterize sucrose viscosity as a function of RH and T.

This parameterization is further combined with observed tropospheric RH and T fields to estimate the viscosity (and phase state) of organic aerosol particles throughout the troposphere. Additionally, the authors use these viscosity distributions to assess  $N_2O_5$  uptake on organic aerosols.

Overall, the manuscript addresses an important and timely topic that aligns well with the scope of the journal. However, several key assumptions are insufficiently discussed, and there are minor issues that require attention (see specific comments below). Therefore, I recommend major revisions before the manuscript can be considered for publication.

**General comments:**

C1: The authors use re-analysis data to calculate the average RH and T fields in the troposphere, and then use these RH and T-distributions to estimate the phase state. However, the source of these RH and T data and their representativeness for atmospheric conditions require further clarification in the main text, as this is somewhat hidden on L68-73 of the SI. It also remains unclear why a climatological dataset was not considered. A more detailed discussion of the rationale and potential implications of this choice would strengthen the manuscript.

C2: The authors provide novel data on the T-dependence of sucrose viscosity and use this data to derive a fragility parameter of 13. The way this fragility parameter is derived and how the determined fragility value compares to previous estimates of the fragility parameter is largely lacking and an in-depth discussion thereof should be added to the revised manuscript, as the estimated viscosity data is sensitive to the fragility value assumed fragility parameter. In addition, the authors claim their fragility parameter to be the "first experimentally constrained estimate of Df (L132). It could be valuable to provide a similar analysis as provided in their Fig. 3, but calculated for a fragility parameter of 10, as traditionally assumed in several studies, to clarify the role of this new, experimentally better constrained fragility parameter.

C3: The authors use sucrose as a proxy of atmospheric organic aerosols. However, the discussion as to what extent the viscosities measured for sucrose particles can be extrapolated to complex atmospheric organic aerosols is insufficient and should be extended.

C4: There are several assumptions made in estimating the effect of particle viscosity on the N2O5 uptake in Sect. 2.5 that require further explanation and/or justification (e.g., choice of numeric values for various parameters in eqs. 3 and 4). As the effect on N2O5 uptake is prominently advertised in the manuscript title, this Section will need to be much improved.

C5: I am unclear about the choice of the content of Sect. 4 on the atmospheric implications. Why are only the estimates on the N2O5 uptake included here and not also the content of the current Sect. 3.2? Combining these two Sections (3.2 and 4.1) into one "Atmospheric implications" section and providing some further discussion on possible caveats on extrapolating their measurements to atmospheric conditions and particles would be helpful.

**Specific comments:**

L31-33: There are plenty of more recent (review) articles for this statement and much work has been done since the publication of the cited works. Please consider adding some more recent references to this statement, such as, but not limited to, e.g. (Zhang et al., 2021).

L35: It could be meaningful to specify as "amorphous semi-solid" and "amorphous solid (or glassy)", at least here when you first introduce the different phase states, to avoid confusion with crystalline solid phase states that OA can also exist in.

L36: Consider introducing the symbol  $(\eta)$  used to represent dynamic viscosity already here and then use it throughout.

L38-39: "Accurate determination of aerosol viscosity and phase state...." Here and elsewhere (e.g., L50) it is unclear to me if you use "viscosity" and "phase state" synonymously or not. Please clarify.

L45: Consider adding (Shiraiwa et al., 2017)

L50: Please add (Koop et al., 2011)

L54: Please add (Petters et al., 2019)

L57-63: You might find the following helpful: (Li et al., 2020; Li and Knopf, 2021)

L60: Please specify what you mean with "reactivity"

L66: I think the year for the Bateman study should be 2015, please check: <a href="https://pubs.acs.org/doi/10.1021/jp508521c">https://pubs.acs.org/doi/10.1021/jp508521c</a>

L82: Renbaum-Wolff et al. 2013a and Renbaum-Wolff et al. 2013b refer to the same paper, please correct.

L89: What do you mean with "thermodynamic conditioning"? Please specify. (Also on L103)

Fig. S1: Caption: Red line is not dotted, but solid, please fix. Please also add information on the goodness of fit to the figure, e.g., RMSE and R2.

Fig. S2: The colored position labels appear very blurry and unreadable in most of the images. Please fix.

Fig. S3a/b:

- Caption: "... the uncertainty in RH from the RH sensor during calibration at each temperature..." I do not follow this. Why is the uncertainty in RH not the same across all data points? Is this uncertainty not given by the error/uncertainty in your RH-sensor? Please specify in the text how the RH uncertainty was determined more clearly by giving a reference to Section S1, where your deliquescence values are described.

- Why is there only a single data point for 273 K?

L110: Fig. S4: I cannot really see an evolution of the particle morphology in these panels. For each RH/T combination only 1 image before and 1 image after poking is shown. It could help to add one more column of images, where the particle is shown after a fixed time after poking (e.g. 10 s) for ALL RH/T combinations. Right now, each RH/T combination shows a different experimental flow time, which makes a comparison very hard.

Fig. S5: Caption. Add "." After Sellier et al. (2015).

L119: Specify: "to temperature changes"

L121: Here and in some other equations the formatting of "In" is off, as it is italicized in some eqs. and not in others. Cf. dominator vs. denominator in eq. (2). Please fix.

L123: What is "infinite temperature"? Please also move the reference to Angell 1991 in front of the comma.

L125: Why have a "(293K)" in the nominator?

L129-133: It is unclear how you determined a fragility parameter from your Fig. S8. A detailed description of the steps how the experimental data was used to determine the fragility parameter should be added to the SI of this work. Looking over the work by Derieux et al. (2018) cited by the authors, it appears to me that they suggest a "lower limit fragility parameter of  $10 \pm 1.7$ ". This does not seem too far off from the reported value of  $13 \pm 1$ . A more detailed description of what the fragility parameter means for the interpretation of your data and those from previous work should be added to better rationalize differences between the fragility parameter reported by you and others.

L134: Please replace "Simulation" with "Estimation of" here and elsewhere

L135: Here and elsewhere, please check placement of references and punctuation.

Sect. 2.5: The various assumptions made in this section should be explained in more detail. It is unclear to me how the chosen parameters fit to your experimental conditions. For example, the authors use a coefficient for the surface reaction of 2.5 x  $10^{-4}$  bas on the "low RH conditions" by Grzinic et al. 2015. Are these low RH conditions representative for the entire RH-range probed by you? Similar, an explanation should be given, why  $\alpha_b = 0.035$  can be chosen.

L154: "The result shows..." This statement is misleading. You might observe viscosity changes over 9 orders of magnitude when comparing across the RH range from ~10-90%, but for any given RH value, the range of viscosities observed across the different temperatures tested is much less than 9 orders or magnitude. Please reformulate more carefully.

Fig. 1: Caption: It is unclear to me, how the colored, dashed fit-lines were established. Are these fits constrained to the viscosity of water at 100 % RH? Is this a fit to just the data points of this study or are literature values taken into account where available, i.e., at 293 K?

L164: "..., indicating a semi-solid state." I think the reasoning or cause-effect relationship is off here. The fact that the reduction in viscosity at 50% RH was lager when going from 273 K 303 K as compared to at 80% RH does not allow you to make statements about the phase state. I.e., it is not the change in viscosity but the absolute viscosity values that allow you to make statement about the aerosol phase state. Please improve the description. Maybe I am misinterpreting what you want to say here?

L165: Please see my comment above. "Flow time contrast" for what? To fully recover to the initial particle morphology after poking? Please clarify.

L170: Should the viscosity threshold for a "glassy" phase state not be > 1012 Pa s?

L185: "Figure 2..." This sentence is incomplete, please fix.

L201: "Viscosity isopleths..." This sentence is incomplete, please fix.

L183: It could be helpful to add a reference to your Section S2 and/or your eq. (S2.1) here, to clarify how the RH and T-dependent viscosity values were derived.

L188: "...the contours are nearly vertical". While true, it could be more appropriate to talk about a slope here, as you do further down. To be more quantitative, you could consider discussing the slope of the black, dashed line (semi-solid/liquid transition) across the 3 temperature ranges that you discuss between L187-193.

L207: Add "... we applied the viscosity parameterization (eq. S2.1)..."

L207-208: Do you refer to the RH and T fields shown in Fig. S9? If yes, a reference would be appropriate. The authors should also add further information about where this RH and T-information was retrieved from. It is stated that this is re-analysis data. Re-analysis of what data? The authors should also clarify why only a period of 2020-2024 was used to determine the average RH and T fields and why they consider this representative?

L209-201: "... derived from monthly mean ambient RH and T ..." Could this not be simplified here in the main text by saying that these are average RH and T profiles over the time period 2020 to 2024? It also seems inconsistent to write about monthly averages here and annual averages on L240.

L211: Add "fractional Stokes-Einstein..."

L218: Please quantify "extremely high RH".

L219: Viscosity is not "suppressed", rather does the presence of water vapor and water in the particles prevent the formation of highly viscous phase states. Please improve formulation.

L225-226: "... together these factors tend ..." I am not sure I follow this sentence. What are multi-component SOA fields?

L230: Can you give some examples of these chemical transport models and provide some references for the assumption of fast mixing?

L233: Add "as a proxy for (secondary?) organic aerosol. However, real..."

L236: I would probably delete "under humid conditions". As most atmospheric inorganic substances undergo deliquescence and efflorescence, it will dependent on the history of the particle, whether the inorganic components of an internally mixed organic-inorganic aerosol are dissolved and hence can promote water uptake or not.

Sect. 4.1: General: It is not entirely clear to me how you use the RH- and T-dependent viscosity data to calculate the N2O5 uptake. Going back to L148-149 it seems like you use the viscosities along with the fractional-Stokes Einstein relationship to calculate N2O5 diffusion coefficients and then use these calculated diffusion coefficients in eqs. (4) and (3), as described in SI L53-5. It would be helpful to make this clearer in the main text, e.g., on L148-149 and maybe also re-iterate at the beginning of Sect. 4.1.

L251: "... could extend over several shorts of the year..." It is unclear what you mean here, please rephrase.

L252: Please quantify "cold conditions"

L257: "uptake rates of" delete "of"

L260: "As shown in Fig. 3c" This sentence seems incomplete and is hard to understand, please reformulate.

L263: For better comparison of your current Fig. 3c and Fig. S11, it would be helpful to have them next to each other. Consider splitting Fig. S11 into two panels, where one panel reproduces your Fig. 3c and the other panel displays the results of your sensitivity analysis.

L266: I would not call this a climatology, which usually would contain data over several decades. Rather use the wording of "average RH- and T-fields" or similar.

L266: I would encourage the authors to tune down the wording here a little bit. There have been previous attempts to explore the phase state of organic aerosol throughout the troposphere, as evident by the studies cited in this work. Related, with this statement the authors seem to claim that the sucrose viscosity is representative for the viscosity of complex atmospheric aerosol, directly contradicting their statement on L236-237.

**Supporting Information:**

L32: Delete "." In front of "(Weight, 2019)"

L36: Move "(Price et al. 2016)" in front of the "."

L58: Add "." After Kiland et al. (2023).

L66: eq. (S2.3) what values of k was used here? It would be helpful to replace "k" by the Greek symbol "kappa" ( $\kappa$ ) here and elsewhere, as traditionally used to describe aerosol hygroscopicity.

L75: "Petters" is misspelled.

L79: Numbering of eq. is off and should read "(S3.1)"

Fig. S7: It is unclear to me how you use eq. (S2.1) to determine a hygroscopicity value from this data. Please specify in the text.

Fig. S9: Please increase the size of this figure over the full page to allow the reader to read out values from the figure. In its current form it is extremely hard to read the figure. The authors could also consider removing the panels of the different seasons, if a season-dependent viscosity is not discussed.

Fig. S10: Please increase the size of this figure over the full page to allow the reader to read out values from the figure. In its current form it is extremely hard to read the figure.

L268: Why do the authors only mention the need of T-dependent viscosity parameterizations? What about the role of RH for organic aerosol viscosity, which according to your Fig. 2 is the key parameter in controlling the aerosol phase state in the troposphere.

**References:**

Koop, T., Bookhold, J., Shiraiwa, M., and Pöschl, U.: Glass transition and phase state of organic compounds: dependency on molecular properties and implications for secondary organic aerosols in the atmosphere, Physical Chemistry Chemical Physics, 13, 19238–19255, https://doi.org/10.1039/C1CP22617G, 2011.

Li, J. and Knopf, D. A.: Representation of Multiphase OH Oxidation of Amorphous Organic Aerosol for Tropospheric Conditions, Environ. Sci. Technol., https://doi.org/10.1021/acs.est.0c07668, 2021.

Li, J., Forrester, S. M., and Knopf, D. A.: Heterogeneous oxidation of amorphous organic aerosol surrogates by O3, NO3, and OH at typical tropospheric temperatures, Atmospheric Chemistry and Physics, 20, 6055–6080, https://doi.org/10.5194/acp-20-6055-2020, 2020.

Petters, S. S., Kreidenweis, S. M., Grieshop, A. P., Ziemann, P. J., and Petters, M. D.: Temperature- and Humidity-Dependent Phase States of Secondary Organic Aerosols, 46, 1005–1013, https://doi.org/10.1029/2018gl080563, 2019.

Shiraiwa, M., Li, Y., Tsimpidi, A. P., Karydis, V. A., Berkemeier, T., Pandis, S. N., Lelieveld, J., Koop, T., and Pöschl, U.: Global distribution of particle phase state in atmospheric secondary organic aerosols, Nature Communications, 8, 15002, https://doi.org/10.1038/ncomms15002, 2017.

Zhang, Y., Liu, P., Han, Y., Li, Y., Chen, Q., Kuwata, M., and Martin, S. T.: Aerosols in Atmospheric Chemistry, American Chemical Society, https://doi.org/10.1021/acsinfocus.7e5020, 2021.

---

## Author Comment (AC1)

Dear Markus Ammann,

Listed below are our responses to the comments from the reviewers of our manuscript. For clarity and visual distinction, the referee comments or questions are listed here in black and are preceded by bracketed, italicized numbers (e.g. [1]). Authors' responses are in red below each referee statement with matching numbers (e.g. [A1]). We thank the reviewers for carefully reading our manuscript and for their very helpful suggestions!

Sincerely,
Mijung Song

**Anonymous Referee #1**

This is the review of the manuscript entitled "Temperature–RH Dependent Viscosity of Organic Aerosols from 273 to 303 K: Implications for Global $N_2O_5$ Uptake" by Ullah et al.

This work measures the viscosity of sucrose–$H_2O$ droplets serving as organic aerosol surrogates using bead mobility and poke-and-flow experiments for temperatures between 273 and 300 K and 20-90% RH. The viscosity values which span 9 orders of magnitude are then applied to estimate the fragility of aqueous sucrose droplets using a fit to the Vogel–Fulcher–Tammann formulation. The experimentally derived viscosity and fitted fragility are used to extend viscosity predictions to 230 K and corresponding characteristic mixing time scales, and N2O5 uptake for typical tropospheric temperatures and humidity. The topic of this study fits well within the scope of the journal Atmospheric Chemistry and Physics. In places, more information needs to be provided to better understand the implications. Measurements are performed over 30 K but then the data is extrapolated over 43 K to cover typical tropospheric conditions. The interpretation of N2O5 uptakes under tropospheric conditions could be ambiguous.

An ACP conclusion section is missing. The current atmospheric implication section seems not to qualify.

General comments:

[1] Sample preparation: The droplets are 40-100 μm in diameter. A conditioning and mixing time scale are given. The former seems to be experimentally set. The latter is derived. A ratio of greater than 1 is taken as evidence that the droplets are in equilibrium with RH. Typically, the indication that solution droplets are in equilibrium with the surrounding RH is when they stopped growing. Since water uptake is proportional to 1/r, growth for such large particles can take long (several minutes, 10s of minutes). A mixing time scale might not be sufficient to assure equilibrium. Conditioning must be longer than the mixing time scale but also longer than the growth process to take up sufficient water. Estimates of these numbers (see textbooks, like Seinfeld and Pandis) would be beneficial to have. Given the conditioning times, I feel that the prerequisite is fulfilled, however, providing a number for the largest droplets would be beneficial to the reader.

[A1] We thank the reviewer for this helpful comment. In the bead-mobility experiments, we used relatively large droplets (typically 40–100 μm in diameter, up to ~100 μm) to track beads reliably, and these measurements were performed at high RH (~60–90%). Before each RH point, droplets were conditioned at the target RH and temperature for 10–30 min before bead tracking, with a conditioning time of 10 min at 303, 293, and 283 K, and 30 min at 273 K.

To assess whether equilibration was sufficient, we calculated the characteristic mixing time of water within the organic matrix ($\tau_{mix,H_2O}$) following the same approach used in prior studies (see Table S1). Across our bead-mobility conditions, the applied conditioning times were typically tens to hundreds of times longer than $\tau_{mix,H_2O}$, indicating that droplets were likely in equilibrium with the surrounding RH at the time of measurement.

To clarify this point for readers, we revised the manuscript to describe the conditioning protocol explicitly. The following text has been added to Section 2.2 (Lines 93–98 in the revised manuscript):

"Prior to each bead-mobility measurement, sucrose-$H_2O$ droplets (diameter ~40–100 μm) were held at the target temperature and RH (> ~60%) with a conditioning time of ~10 min at 303, 293, and 283 K, and ~30 min at 273 K. At each temperature, RH was decreased in increments of ~5%, with droplets conditioned at each RH prior to measurement. Equilibration was further assessed using the characteristic mixing time of water within the droplet matrix ($\tau_{mix,H_2O}$) calculated as

described in Sect. S1. The applied conditioning times were much longer than $\tau_{mix,H_2O}$ (Table S1), indicating that the droplets were held for sufficient time to reach equilibrium prior to measurement."

[2] Viscosity was measured across 30 K but then extrapolated by 43 K to 230 K. This implies a large caveat. Especially considering that at lower temperatures, a very small change in humidity will have a substantial effect on viscosity. Hence such a large extrapolation can yield large uncertainties. There is uncertainty in RH from measurements, from the VFT fit, and from averaging the reanalysis data. This should be clearer articulated in manuscript abstract and conclusions. For these reasons, one could argue that the application to tropospheric conditions may not be similarly emphasized compared to the laboratory viscosity measurements.

[A2] We thank the reviewer for this important point and fully agree that extrapolating viscosity data measured over 273–303 K down to 230 K introduces a substantial caveat. At low temperatures, viscosity becomes extremely sensitive to small changes in RH, such that uncertainties in RH control, the VFT fit, and the use of averaged reanalysis T–RH fields can lead to order-of-magnitude differences in predicted viscosity and phase state. Thus, we have revised the manuscript to clearly articulate these limitations and rebalance the emphasis between laboratory measurements and tropospheric applications in the Abstract and Conclusion.

Abstract:

"When coupled with zonal-mean tropospheric T–RH fields (2020–2024), the parameterization was used to infer global distributions of viscosity and organic-phase mixing time ($\tau_{mix,org}$) for 200 nm particles, suggesting predominantly liquid states below ~2 km, semisolid regimes across ~2–9 km (latitude dependent), and near-glassy conditions at higher altitudes; $\tau_{mix,org}$ was typically < 1 h in the boundary layer but often exceeded 1 h aloft. The resulting global fields highlight the potential atmospheric implications of temperature-sensitive viscosity across the troposphere."

In addition, we have included the Conclusion, which was missing, in the revised manuscript:

"In this study, we experimentally measured the viscosity of sucrose–H$_2$O droplets, used as a proxy for SOA, over a temperature range of 273–303 K and RH of ~20–90 % using bead-mobility and poke-and-flow techniques. Across this combined temperature–RH space, viscosity spanned approximately eight orders of magnitude. At high RH (> ~80 %), droplets exhibited liquid behavior

across the investigated temperature range, with viscosity decreasing modestly as temperature increased from 273 to 303 K. At intermediate RH (~50 %), increasing temperature from 273 to 303 K reduced viscosity by up to three orders of magnitude, while droplets remained highly viscous and exhibited semisolid behavior. At lower RH, droplets showed brittle cracking during poke-and-flow experiments. Notably, cracking occurred at higher RH at lower temperatures, whereas at higher temperatures cracking was observed only at lower RH, with no observable restorative flow once fracture occurred, consistent with viscosities reaching the experimental lower-limit values (~$10^8$ Pa s).

  Based on these experimentally constrained viscosity measurements, a VFT formulation with an experimentally derived fragility parameter ($D_f$ = 13 ± 1) was used to extend the viscosity parameterization across a broader tropospheric temperature–RH space. Application of this framework to zonal-mean tropospheric temperature and RH fields suggested a vertically stratified aerosol phase state, with predominantly liquid behavior near the surface, a transition to semisolid states aloft, and increasingly diffusion-limited conditions toward the upper troposphere. Consistent with this stratification, $\tau_{mix,org}$ was inferred to remain shorter than ~1 h in the boundary layer but to frequently exceed ~1 h at higher altitudes, implying that equilibrium assumptions may not hold under cold and dry tropospheric conditions. As a result, diffusion-limited heterogeneous processes, such as $N_2O_5$ uptake, were inferred to be less efficient aloft than in the boundary layer.

  While uncertainties remain due to extrapolation beyond the experimental temperature range, this study provides experimentally constrained temperature–RH-dependent viscosity data and a physically grounded framework for assessing aerosol phase state, mixing behavior, and diffusion-limited heterogeneous chemistry under tropospheric conditions. Future work should extend viscosity measurements to lower temperatures and RH, incorporate chemically complex organic–inorganic aerosol systems, and explicitly evaluate viscosity-dependent mixing and reactivity in atmospheric models to further reduce uncertainties in multiphase process representations."

[3] More information has to be provided how the altitude–latitude profiles of zonal-mean temperature and RH from Copernicus Data Store were derived. Which ERA5 model was used?
[A3] Thank you for the comment. We have expanded the description of how the latitude–altitude profiles were derived. We used the ECMWF ERA5 reanalysis temperature and RH data obtained from the Copernicus Climate Data Store (ERA5 monthly averaged data on pressure levels) and

averaged the monthly fields over 2020–2024. Zonal-mean profiles were then constructed by averaging across all longitudes at each latitude and pressure level (around lines 146–154 in the revised manuscript).

"For the global analysis of aerosol phase state, we used the temperature and RH fields from the ECMWF ERA5 reanalysis (ERA5 monthly averaged data on pressure levels) obtained from the Copernicus Climate Data Store (Hersbach et al., 2023). Monthly mean fields for 2020–2024 were retrieved, zonally averaged, and converted from pressure coordinates to geometric altitude using the hypsometric equation to derive latitude-altitude profiles of temperature and RH. The resulting latitude–altitude distributions of temperature and RH were then combined with the laboratory viscosity parameterization to construct zonal and global maps of viscosity and phase state. Additional details on data processing and zonal averaging are provided in Supplementary Information Sect. S4. The ERA5 fields averaged over 2020–2024 were used to represent recent tropospheric temperature and RH conditions.

The supplement does not provide any information. What is the meaning of Eq. S3? A hypsometric equation with a lapse rate?

We thank the reviewer for pointing this out. In the revised Supplementary Information, we have now explicitly explained the meaning and purpose of Eq. (S4.1). Equation (S4.1) is the barometric (hypsometric) formulation used to convert pressure levels to geometric altitude (h). It assumes a constant temperature lapse rate and uses standard constants listed in the SI. We use this conversion to present the vertical distributions on an intuitive altitude axis; the underlying ERA5 fields remain pressure-level monthly means. In the revised SI, we have added the following (around lines 102–105):

"To convert pressure-level data to geometric altitude, Eq. (S4.1) represents the hypsometric relationship assuming a constant temperature lapse rate. This formulation is widely used to visualize the pressure-level meteorological data on an altitude coordinate and does not alter the underlying temperature or RH values (Maclean et al., 2021; Kiland et al., 2023)."

**References:**

- Kiland, K. J., Mahrt, F., Peng, L., Nikkho, S., Zaks, J., Crescenzo, G. V., and Bertram, A. K.: Viscosity, glass formation, and mixing times within secondary organic aerosol from

biomass burning phenolics, ACS Earth Space Chem., 7, 1388–1400, https://doi.org/10.1021/acsearthspacechem.3c00039, 2023.

- Maclean, A. M., Li, Y., Crescenzo, G. V., Smith, N. R., Karydis, V. A., Tsimpidi, A. P., Butenhoff, C. L., Faiola, C., Lelieveld, J., Nizkorodov, S. A., Shiraiwa, M., and Bertram, A. K.: Global distribution of the phase state and mixing times within secondary organic aerosol particles in the troposphere based on room-temperature viscosity measurements, ACS Earth Space Chem., 5, 3458–3473, https://doi.org/10.1021/acsearthspacechem.1c00296, 2021.

Did you use this to convert pressure/temperature fields into heights? Why not show pressure levels instead of altitude?

Yes. We used Eq. (S4.1) to convert the ERA5 pressure-level fields to altitude for plotting. We chose altitude rather than pressure to make the vertical structure (in km) easier to interpret in the context of tropospheric phase state and mixing-time regimes.

Anyway, it is not clear which data sets are used, how they are averaged (value range/uncertainty), etc. This may impact some of the interpretations of the particle phase state, leading to inconsistency. Could published climatological means be used instead?

We considered using published long-term climatological means; however, we chose ERA5 because it provides internally consistent RH and T on the same grid and a transparent reanalysis framework. We use the 2020–2024 multi-year mean as a recent climatological window; using a longer climatology would further smooth interannual variability but is not expected to change the qualitative latitude–altitude patterns emphasized here.

We have added the following text in SI Section S4 (around lines 97–101 in the revised SI):

"Monthly mean temperature and RH on pressure levels were obtained from the ERA5 reanalysis (Copernicus Climate Data Store) for 2020–2024. Latitude–altitude profiles were constructed by averaging across all longitudes at each latitude and pressure level. Pressure levels were converted to geometric altitude using the barometric (hypsometric) relationship (Eq. S4.1), and the resulting zonal-mean profiles are shown in Fig. S9."

[4] RH impacts the phase state as calculated but only if the particle is in equilibrium with RH. Only then does condensed-phase water activity equals RH. At cold temperatures, when diffusion slows down and the particle viscosity is likely higher, this may not be the case. Has this been considered when looking at zonal plots? I assume, even at colder temperatures the authors assume equilibrium between condensed phase and gas phase. This would be another large caveat. We know that organic particles can be in disequilibrium with the gas phase, e.g., (Berkemeier et al., 2014).

[A4] We thank the reviewer for this important point. Our zonal and global phase-state maps are derived assuming local thermodynamic equilibrium between particle water activity and ambient RH. We agree that this assumption becomes increasingly uncertain under cold, high-altitude conditions, where high viscosity and slow diffusion can kinetically limit gas–particle equilibration (e.g., Berkemeier et al., 2014).

To address the Reviewer's comment, we have now clarified in the revised manuscript that the presented maps represent equilibrium-based reference states inferred from temperature and RH. Under cold and dry conditions, actual particle water content and phase state may deviate from these equilibrium predictions due to kinetic limitations, and the maps should be interpreted accordingly. We have added the following text to the section describing the zonal and global maps: "Under these conditions, sucrose-$H_2O$ droplets may not remain in equilibrium with ambient RH because slow water diffusion can prevent condensed-phase water activity from equilibrating with gas-phase RH, particularly at low temperatures and high viscosities (Berkemeier et al., 2014)." (Lines 256–258)

**Reference:**

- Berkemeier, T., Shiraiwa, M., Pöschl, U., and Koop, T.: Competition between water uptake and ice nucleation by glassy organic aerosol particles, Atmos. Chem. Phys., 14, 12513–12531, https://doi.org/10.5194/acp-14-12513-2014, 2014.

[5] The greater extent of liquid and semisolid particles at higher altitudes over the southern polar region compared to the extra tropics and tropics seems to be counterintuitive. At surface levels maybe, but at heights above 1-2 km, one would expect the particle phase to be on the solid side due to extremely low temperatures. The resulting greater N2O5 uptake in this region is not discussed.

[A5] Thank you for this comment. We agree that the greater extent of liquid and semisolid particles at higher altitudes over the southern polar region may appear counterintuitive if temperature alone is considered. However, as shown in Fig. S9 (panels i and j), the southern polar free troposphere is characterized by persistently high RH, which counteracts the effect of low temperatures by enhancing water uptake and plasticization of the organic phase.

As a result, the transition to highly viscous or glassy states is delayed relative to drier extratropical and tropical regions, allowing particles to remain partially liquid or semisolid over a broader altitude range. This RH-driven reduction in viscosity supports continued molecular diffusion and helps explain the relatively higher $N_2O_5$ uptake predicted over the southern polar region despite the low temperatures. Corresponding clarification has been added to the revised manuscript (Lines 242–246).

"At high southern latitudes (> 70°S), liquid phases extended to ~3–4 km above the surface (Fig. 3a), higher than in the tropics and northern polar regions, where liquid states were typically confined below ~2 km. This persistence was consistent with the high RH (> 80%) observed in these regions (Fig. S9), which enhanced water uptake and plasticization of the organic matrix, thereby delaying the formation of highly viscous or glassy phase states."

**Specific comments:**

[6] Line 32-33: Please update with more recent reviews or articles on these topics.

[A6] As suggested, the following references have been added to the revised manuscript (Line 32–34).

- Bei, N., Xiao, B., Wang, R., Yang, Y., Liu, L., Han, Y., and Li, G.: Impacts of aerosol–radiation and aerosol–cloud interactions on a short-term heavy-rainfall event–a case study in the Guanzhong Basin, China, Atmos. Chem. Phys., 25, 10931–10948, https://doi.org/10.5194/acp-25-10931-2025, 2025.

- El Haddad, I., Vienneau, D., Daellenbach, K. R., Modini, R., Slowik, J. G., Upadhyay, A., Vasilakos, P. N., Bell, D., de Hoogh, K., and Prevot, A. S. H.: Opinion: How will advances in aerosol science inform our understanding of the health impacts of outdoor particulate pollution?, Atmos. Chem. Phys., 24, 11981–12011, https://doi.org/10.5194/acp-24-11981-2024, 2024.

- Manavi, S. E. I., Aktypis, A., Siouti, E., Skyllakou, K., Myriokefalitakis, S., Kanakidou, M., and Pandis, S. N.: Atmospheric aerosol spatial variability: Impacts on air quality and climate change, One Earth, 8, https://doi.org/10.1016/j.oneear.2025.101237, 2025.

- McNeill, V. F.: Atmospheric aerosols: clouds, chemistry, and climate, Annu. Rev. Chem. Biomol. Eng., 8, 427–444, https://doi.org/10.1146/annurev-chembioeng-060816-101538, 2017.

- Nault, B. A., Jo, D. S., McDonald, B. C., Campuzano-Jost, P., Day, D. A., Hu, W. W., Schroder, J. C., Allan, J., Blake, D. R., Canagaratna, M. R., Coe, H., Coggon, M. M., DeCarlo, P. F., Diskin, G. S., Dunmore, R., Flocke, F., Fried, A., Gilman, J. B., Gkatzelis, G., Hamilton, J. F., Hanisco, T. F., Hayes, P. L., Henze, D. K., Hodzic, A., Hopkins, J., Hu, M., Huey, L. G., Jobson, B. T., Kuster, W. C., Lewis, A., Li, M., Liao, J., Nawaz, M. O., Pollack, I. B., Peischl, J., Rappenglück, B., Reeves, C. E., Richter, D., Roberts, J. M., Ryerson, T. B., Shao, M., Sommers, J. M., Walega, J., Warneke, C., Weibring, P., Wolfe, G. M., Young, D. E., Yuan, B., Zhang, Q., de Gouw, J. A., and Jimenez, J. L.: Secondary organic aerosols from anthropogenic volatile organic compounds contribute substantially to air pollution mortality, Atmos. Chem. Phys., 21, 11201–11224, https://doi.org/10.5194/acp-21-11201-2021, 2021.

- Su, H., Cheng, Y. F., and Pöschl, U.: New multiphase chemical processes influencing atmospheric aerosols, air quality, and climate in the Anthropocene, Acc. Chem. Res., 53, 2034–2043, https://doi.org/10.1021/acs.accounts.0c00246, 2020.
- Sun, Y. L., Luo, H., Li, Y., Zhou, W., Xu, W. Q., Fu, P. Q., and Zhao, D. F.: Atmospheric organic aerosols: online molecular characterization and environmental impacts, npj Clim. Atmos. Sci., 8, https://doi.org/10.1038/s41612-025-01199-2, 2025.
- Wall, C. J., Norris, J. R., Possner, A., Mccoy, D. T., Mccoy, I. L., and Lutsko, N. J.: Assessing effective radiative forcing from aerosol-cloud interactions over the global ocean, Proc. Natl. Acad. Sci. U.S.A., 119, https://doi.org/10.1073/pnas.2210481119, 2022.
- Zhang, Y., Liu, P., Han, Y., Li, Y., Chen, Q., Kuwata, M., and Martin, S. T.: Aerosols in Atmospheric Chemistry, ACS In Focus, American Chemical Society, https://doi.org/10.1021/acsinfocus.7e5020, 2021

[7] Line 39-40, and 57: The following works who studied uptake kinetics for RH variation and temperatures of the upper troposphere could be cited as well: (Li and Knopf, 2021; Li et al., 2020; Slade and Knopf, 2014).

[A7] The suggested references have now been incorporated into the revised manuscript:

- Li, J. N. and Knopf, D. A.: Representation of multiphase OH oxidation of amorphous organic aerosol for tropospheric conditions, Environ. Sci. Technol., 55, 7266–7275, https://doi.org/10.1021/acs.est.0c07668, 2021.
- Li, J. N., Forrester, S. M., and Knopf, D. A.: Heterogeneous oxidation of amorphous organic aerosol surrogates by O3, NO, and OH at typical tropospheric temperatures, Atmos. Chem. Phys., 20, 6055–6080, https://doi.org/10.5194/acp-20-6055-2020, 2020
- Slade, J. H. and Knopf, D. A.: Multiphase OH oxidation kinetics of organic aerosol: The role of particle phase state and relative humidity, Geophys. Res. Lett., 41, 5297–5306, https://doi.org/10.1002/2014gl060582, 2014.

[8] Line 44 and following: Study by (Lienhard et al., 2015) is missing. They also parameterized water diffusion in sucrose for different water activities.

[A8] The suggested reference has now been incorporated into the revised manuscript:

- Lienhard, D. M., Huisman, A. J., Krieger, U. K., Rudich, Y., Marcolli, C., Luo, B. P., Bones, D. L., Reid, J. P., Lambe, A. T., Canagaratna, M. R., Davidovits, P., Onasch, T. B., Worsnop, D. R., Steimer, S. S., Koop, T., and Peter, T.: Viscous organic aerosol particles in the upper troposphere: diffusivity-controlled water uptake and ice nucleation, Atmos. Chem. Phys., 15, 13599–13613, https://doi.org/10.5194/acp-15-13599-2015, 2015.

[9] Line 50-51: The article by (Knopf et al., 2024) would be beneficial in this list. They also discuss uptake at low temperatures.

[A9] The suggested reference has now been incorporated into the revised manuscript:

- Knopf, D. A., Ammann, M., Berkemeier, T., Pöschl, U., and Shiraiwa, M.: Desorption lifetimes and activation energies influencing gas-surface interactions and multiphase chemical kinetics, Atmos. Chem. Phys., 24, 3445–3528, https://doi.org/10.5194/acp-24-3445-2024, 2024.

[10] Line 59: Surfactants and phase separation can also reduce the uptake of N2O5. See, e.g., (McNeill et al., 2006; Cosman et al., 2008; Gaston et al., 2014).

[A10] The suggested references have now been incorporated into the revised manuscript:

- Cosman, L. M., Knopf, D. A., and Bertram, A. K.: N2O5 reactive uptake on aqueous sulfuric acid solutions coated with branched and straight-chain insoluble organic surfactants, J. Phys. Chem. A, 112, 2386–2396, https://doi.org/10.1021/jp710685r, 2008.
- Gaston, C. J., Thornton, J. A., and Ng, N. L.: Reactive uptake of N2O5 to internally mixed inorganic and organic particles: the role of organic carbon oxidation state and inferred organic phase separations, Atmos. Chem. Phys., 14, 5693–5707, https://doi.org/10.5194/acp-14-5693-2014, 2014.
- McNeill, V. F., Patterson, J., Wolfe, G. M., and Thornton, J. A.: The effect of varying levels of surfactant on the reactive uptake of N2O5 to aqueous aerosol, Atmos. Chem. Phys., 6, 1635–1644, https://doi.org/10.5194/acp-6-1635-2006, 2006
- Ryder, O. S., Campbell, N. R., Morris, H., Forestieri, S., Ruppel, M. J., Cappa, C., Tivanski, A., Prather, K., and Bertram, T. H.: Role of organic coatings in regulating N2O5 reactive uptake to sea spray aerosol, J. Phys. Chem. A, 119, 11683–11692, https://doi.org/10.1021/acs.jpca.5b08892, 2015

After incorporating the additional references, the corresponding lines (62–65) in the revised manuscript have been updated as follows:

"Modeling and observations have indicated that the presence of viscous organic shells or surfactant-rich surface layers can reduce the $N_2O_5$ uptake coefficient compared to liquid particles (McNeill et al., 2006; Cosman et al., 2008; Gaston et al., 2014; Ryder et al., 2015; Song et al., 2025)."

[11] Line 123-124: Please elaborate what is meant by infinite viscosity? The viscosity at which glass transition manifests?

[A11] In Eq. (1) and (2), $\eta_\infty$ is not a physically measured viscosity at a real 'infinite temperature', but the pre-exponential factor in the Vogel–Fulcher–Tammann relation, corresponding to the extrapolated high-temperature limiting viscosity ($T \rightarrow \infty$). Following previous aerosol studies, we fix $\eta_\infty$ to $1 \times 10^{-5}$ Pa·s. To make the sentence more consistent and clearer, we have revised the text to state that $\eta_\infty$ is the 'high temperature limiting viscosity (pre-exponential factor)' rather than 'viscosity at infinite temperature.

The following text has been revised in the manuscript (around lines 131–134):

"$\eta(RH, T)$ is the viscosity at a given RH and temperature, $\eta_\infty = 1 \times 10^{-5}$ Pa·s represents the high-temperature limiting viscosity in the VFT formulation (Angell, 1991). This parameter reflects the liquid-like behavior of the material at high temperature and should be distinguished from the glass-transition viscosity, typically $\sim 10^{12}$ Pa·s."

[12] Line 128: How do you derive kappa from previous equations? No relationship with kappa is given.

[A12] In our analysis, the hygroscopicity parameter $\kappa$ is not introduced independently but is retrieved as a fitting parameter within the viscosity parameterization. Specifically, viscosity at 293 K is described using a mole-fraction-based Arrhenius mixing rule, in which the organic mole fraction ($x_{org}$) is derived from the organic mass fraction ($w_{org}$). The dependence of $\kappa$ enters through $w_{org}$, which is related to water activity ($a_w = RH/100$) using the conventional $\kappa$-based mixing relationship (Eq. S2.3). By fitting this combined Arrhenius–$\kappa$ formulation to the measured viscosity–RH data at 293 K, $\kappa$ is obtained. We have clarified this relationship in the main text (around lines 135–138) and Sect. S2 of the Supplementary Information.

"where $\eta(RH, 293\ K)$ is the viscosity at 293 K, estimated using a mole-fraction-based Arrhenius mixing rule in which the organic mole fraction ($x_{org}$) is obtained from water activity ($a_w$) via a mass-based hygroscopicity parameter $\kappa$ using Eq. (S2.3). Fitting this formulation to the measured viscosity–RH data at 293 K yields $\kappa = 0.061 \pm 0.0023$ (Fig. S7); details of the parametrization are provided in Sect. S2. "

[13] Line 147-149: For the reacto-diffusive length, did you account for the temperature and humidity dependency for the N2O5 diffusion coefficient and for the rate constant? Maybe a supplemental plot of this length as a function of temperature and humidity could be beneficial for the reader.

[A13] In this work, we follow the same reacto-diffusive framework and parameter assumptions adopted in Song et al. (2025), in which $N_2O_5$ uptake is described using a resistor-model formulation, with the hydrolysis rate constant fixed at $k^I = 4.1\ ns^{-1}$, based on molecular simulation results by Galib and Limmer (2021). As in Song et al. (2025) potential temperature or RH dependencies of $k^I$ are not explicitly included.

The key extension in the present study is that the $N_2O_5$ diffusivity is derived from a temperature- and RH-dependent viscosity parameterization using the VFT equation and the fractional Stokes–Einstein relation. Consequently, while the underlying physical framework and reaction assumptions are consistent with Song et al. (2025), the temperature and RH dependence of the reacto-diffusive length arises here from the explicitly constrained viscosity, rather than from assumed or fixed diffusivity values.

**References:**
- Galib, M. and Limmer, D. T.: Reactive uptake of N2O5 by atmospheric aerosol is dominated by interfacial processes, Science, 371, https://doi.org/10.1126/science.abd7716, 2021.
- Song, M., Li, Y., Seong, C., Yang, H., Jang, K.-S., Wu, Z., Lee, J. Y., Matsuki, A., and Ahn, J.: Direct observation of liquid–liquid phase separation and core–shell morphology of PM2.5 collected from three northeast Asian cities and implications for N2O5 hydrolysis, ACS ES&T Air, 2, 1079–1088, https://doi.org/10.1021/acsestair.5c00043, 2025

[14] Line 175, figure 1: In caption, please state the component investigated.

[A14] The investigated compound name has been added to the caption.

"Viscosity data for sucrose-$H_2O$ droplets obtained from bead mobility and poke-and-flow experiments at $273 \pm 1$ K (magenta circles), $283 \pm 1$ K (blue circles), $293 \pm 1$ K (red circles), and $303 \pm 1$ K (green circles) across an RH range of 20–90%."

[15] Line 183: Viscosity was measured across 30 K but then extrapolated by 43 K. This needs to be strongly pointed out in the abstract and throughout the manuscript. As it will be later (line 198) where it is stated that at lower temperatures a very small change in humidity will have a substantial effect on viscosity. Hence such a large extrapolation can yield large uncertainties.

[A15] Please see our response to [A2].

[16] Line 194-197: This reads like a repetition of previous paragraph?

[A16] The repeated sentence has been removed, and the overall flow has been improved in the revised manuscript (around lines 216–220).

"Between 240 and 280 K (typical of the middle troposphere, ~3–8 km), the isopleths display more moderate slopes, indicating a combined influence of both temperature and RH on viscosity and phase state. At higher temperatures (280–310 K, representative of the lower troposphere at ~0–3 km), the isopleths and the phase boundary show gentler slopes, consistent with reduced RH sensitivity and a comparatively stronger temperature control on viscosity."

[17] Line 208-210: How are the Copernicus Climate Data applied to get altitude resolved temperature and humidity levels?

[A17] See our response to [A3].

[18] Line 214-221: This interpretation is based on RH in polar regions. However, polar regions are much colder than the tropics. Maybe this has validity at surface levels, though in winter at 230 K at the surface, particles are likely solid independently of RH. At higher altitudes, it is even less likely that particles are liquid. Is condensed-phase water activity in equilibrium with RH?

[A18] See our response to [A4].

[19] Line 246, section 4.1: You do not discuss the high $N_2O_5$ uptake at greater altitudes in the southern polar region. But state on lines 252-254 that cold conditions reduce uptake. Somehow, there is some inconsistency.

[A19] See our response to [A5].

**Technical corrections:**

[20] Throughout document and supplement: Variables in equations are typically italic font and subscript is in normal font.

[A20] We have corrected this throughout the manuscript.

[21] Line 128 and other instances: Hygroscopicity parameter is typically a Greek kappa.

[A21] We have corrected this throughout the manuscript.

[22] Line 162: Erroneous hyphen between "magnitude" and "from"?

[A22] The unnecessary hyphen has been removed.

[23] Line 251: What do you mean with "several shorts"?

[A23] The sentence has been revised as follows (lines 269–270):

"Using our parameterized viscosity fields, we estimated that in the midlatitude upper troposphere, suppressed uptake could extend over several short periods, particularly in winter. "

[24] Line 260: You mean the "uptake coefficient levels at ….".

[A24] The sentence has been revised as follows (lines 278–280):

"As shown in Fig. 3c, $N_2O_5$ uptake coefficient is leveled off near $\sim 10^{-3.5}$ at higher altitudes, because the surface reaction term ($\mathit{\Gamma}_s$) was fixed at $2.5 \times 10^{-4}$ in Eq. (S5.1)."

**Supplement:**

[25] Line 36: Erroneous period, missing space?

[A25] The misplaced period has been corrected.

[26] Line 37: Missing space before"(Haynes".

[A26] It has been corrected.

[27] Line 137: Superfluous space.

[A27] The superfluous space has been removed.

**Anonymous Referee #2**

The manuscript by Ullah et al. reports new measurements of aerosol dynamic viscosity. Viscosity was determined for particles composed of sucrose–water, a commonly used proxy for atmospheric organic aerosols, at discrete relative humidity (RH) and temperature (T) conditions using established techniques. Notably, the measurements of sucrose particle viscosity as a function of temperature are novel and represent a valuable contribution to literature. The authors apply their experimental data within the Vogel–Fulcher–Tammann (VFT) framework and the Arrhenius mixing rule to parameterize sucrose viscosity as a function of RH and T.

This parameterization is further combined with observed tropospheric RH and T fields to estimate the viscosity (and phase state) of organic aerosol particles throughout the troposphere. Additionally, the authors use these viscosity distributions to assess $N_2O_5$ uptake on organic aerosols.

Overall, the manuscript addresses an important and timely topic that aligns well with the scope of the journal. However, several key assumptions are insufficiently discussed, and there are minor issues that require attention (see specific comments below). Therefore, I recommend major revisions before the manuscript can be considered for publication.

**General comments:**

[1] The authors use re-analysis data to calculate the average RH and T fields in the troposphere, and then use these RH and T-distributions to estimate the phase state. However, the source of these RH and T data and their representativeness for atmospheric conditions require further clarification in the main text, as this is somewhat hidden on L68-73 of the SI. It also remains unclear why a climatological dataset was not considered. A more detailed discussion of the rationale and potential implications of this choice would strengthen the manuscript.

[A1] We thank Reviewer 2 for this helpful comment. As addressed in our response to Reviewer 1 [3], the RH and temperature fields used for the zonal/global phase-state analysis are monthly mean ECMWF ERA5 pressure-level reanalysis data (Copernicus Climate Data Store), averaged over 2020–2024, zonally averaged across longitudes, and converted to geometric altitude using the hypsometric relationship (Eq. S4.1). To reduce repetition and improve transparency, we now describe this dataset (including the rationale for using the 2020–2024 multi-year mean rather than a longer climatology) in a newly added Sect. 2.5 of the main text (lines 146–154), while retaining additional technical details in SI Sect. S4. The resulting maps should be interpreted as equilibrium-based estimates typical of this averaging window.

[2] The authors provide novel data on the T-dependence of sucrose viscosity and use this data to derive a fragility parameter of 13. The way this fragility parameter is derived and how the determined fragility value compares to previous estimates of the fragility parameter is largely lacking and an in-depth discussion thereof should be added to the revised manuscript, as the estimated viscosity data is sensitive to the fragility value assumed fragility parameter. In addition, the authors claim their fragility parameter to be the "first experimentally constrained estimate of Df (L132). It could be valuable to provide a similar analysis as provided in their Fig. 3, but calculated for a fragility parameter of 10, as traditionally assumed in several studies, to clarify the role of this new, experimentally better constrained fragility parameter.

[A2] We thank the reviewer for this constructive comment. In the revised manuscript, we have clarified how the fragility parameter ($D_f$) is derived and how sensitive our results are to its assumed value. As described in the revised Sect. 2.4 and in the newly added Supplementary Information (Sect. S3), all measured viscosities spanning 273–303 K and ~20–90% RH were fitted simultaneously using a global nonlinear least-squares approach to obtain a single, RH-independent

fragility parameter of $D_f = 13 \pm 1$. This approach contrasts with previous studies, in which $D_f$ was typically assumed (commonly $D_f = 10$) rather than experimentally constrained across a broad T–RH space.

To directly address the reviewer's concern regarding sensitivity, we have added a dedicated sensitivity analysis comparing predictions obtained with $D_f = 13 \pm 1$ and with $D_f = 10$, which has been widely adopted as an assumed value in several earlier studies. As shown in the newly added Fig. S12, the resulting global distributions of viscosity and organic-phase mixing time are broadly similar, indicating that the large-scale latitude–altitude structure of aerosol phase state is robust to reasonable variations in $D_f$. The following text has been added to the revised manuscript:

Results & Discussion (Lines: 263–265):

"Sensitivity tests using an alternative $D_f = 10$, which is widely used as an assumed value in previous studies (Maclean et al., 2021b; Kiland et al., 2023; Gerrebos et al., 2025), produced broadly similar latitude–altitude patterns of viscosity and $\tau_{mix,org}$ (Fig. S12)."

**References:**

- Gerrebos, N. G., Browning, L. P., Nikkho, S., Chartrand, E. R., Zaks, J., Wu, C., and Bertram, A. K.: Two-phase morphology and drastic viscosity changes in biomass burning organic aerosol after hydroxyl radical aging, Environ. Sci. Atmos., https://doi.org/10.1039/d5ea00084j, 2025.

- Kiland, K. J., Mahrt, F., Peng, L., Nikkho, S., Zaks, J., Crescenzo, G. V., and Bertram, A. K.: Viscosity, glass formation, and mixing times within secondary organic aerosol from biomass burning phenolics, ACS Earth and Space Chem., 7, 1388–1400, https://doi.org/10.1021/acsearthspacechem.3c00039, 2023.

- Maclean, A. M., Li, Y., Crescenzo, G. V., Smith, N. R., Karydis, V. A., Tsimpidi, A. P., Butenhoff, C. L., Faiola, C. L., Lelieveld, J., Nizkorodov, S. A., Shiraiwa, M., and Bertram, A. K.: Global distribution of the phase state and mixing times within secondary organic aerosol particles in the troposphere based on room-temperature viscosity measurements, ACS Earth and Space Chem., 5, 3458–3473, https://doi.org/10.1021/acsearthspacechem.1c00296, 2021.

[Figure]

Figure S12: Influence of the fragility parameter $D_f$ on the viscosity of sucrose-$H_2O$ and related mixing-times in the troposphere. Panel (a) presents the viscosity of sucrose-$H_2O$ and the mixing times calculated using $D_f = 13$, whereas panel (b) shows the corresponding profiles obtained with $D_f = 10$. All fields are shown as functions of altitude and latitude, derived from annual zonal-mean relative humidity and temperature for the years 2020–2024, obtained from the Copernicus Climate Data Store (https://cds.climate.copernicus.eu/).

[3] The authors use sucrose as a proxy of atmospheric organic aerosols. However, the discussion as to what extent the viscosities measured for sucrose particles can be extrapolated to complex atmospheric organic aerosols is insufficient and should be extended.

[A3] Sucrose-$H_2O$ particles have been extensively used in prior studies investigating RH-dependent viscosity, diffusion coefficients, mixing times, and phase state of viscous or highly oxidized SOA (Zobrist et al., 2011; Power et al., 2013; Bateman et al., 2015; Chenyakin et al., 2017; Maclean et al., 2017; Kiland et al., 2019; Song et al., 2021). These studies support the use

of sucrose as a well-characterized reference system that captures key physical features of highly oxidized, hydrophilic OA, particularly strong water plasticization and glassy behavior at low temperature and RH. To address the reviewer's comment, we have clarified why sucrose has been widely used in previous laboratory and modeling studies and have delineated the scope and limitations of extrapolating sucrose-based viscosity results to atmospheric OA.

Introduction (Lines: 69–71):

"In this study, sucrose was used as a model compound that has been widely employed as a laboratory proxy for SOA, particularly in investigations of aerosol viscosity, diffusion, and phase state (Zobrist et al., 2011; Power et al., 2013; Bateman et al., 2015; Chenyakin et al., 2017; Maclean et al., 2017; Kiland et al., 2019; Song et al., 2021)."

**References:**

- Bateman, A. P., Bertram, A. K., and Martin, S. T.: Hygroscopic influence on the semisolid-to-liquid transition of secondary organic materials, J. Phys. Chem. A, 119, 4386–4395, https://doi.org/10.1021/jp508521c, 2015.

- Chenyakin, Y., Ullmann, D. A., Evoy, E., Renbaum-Wolff, L., Kamal, S., and Bertram, A. K.: Diffusion coefficients of organic molecules in sucrose–water solutions and comparison with Stokes–Einstein predictions, Atmos. Chem. Phys., 17, 2423–2435, https://doi.org/10.5194/acp-17-2423-2017, 2017.

- Kiland, K. J., Maclean, A. M., Kamal, S., and Bertram, A. K.: Diffusion of organic molecules as a function of temperature in a sucrose matrix (a proxy for secondary organic aerosol), J. Phys. Chem. Lett., 10, 5902–5908, https://doi.org/10.1021/acs.jpclett.9b02182, 2019.

- Maclean, A. M., Butenhoff, C. L., Grayson, J. W., Barsanti, K., Jimenez, J. L., and Bertram, A. K.: Mixing times of organic molecules within secondary organic aerosol particles: a global planetary boundary layer perspective, Atmos. Chem. Phys., 17, 13037–13048, https://doi.org/10.5194/acp-17-13037-2017, 2017.

- Power, R. M., Simpson, S. H., Reid, J. P., and Hudson, A. J.: The transition from liquid to solid-like behaviour in ultrahigh viscosity aerosol particles, Chem. Sci., 4, 2597–2604, https://doi.org/10.1039/c3sc50682g, 2013.

- Song, Y.-C., Lilek, J., Lee, J. B., Chan, M. N., Wu, Z., Zuend, A., and Song, M.: Viscosity and phase state of aerosol particles consisting of sucrose mixed with inorganic salts, Atmos. Chem. Phys., 21, 10215–10228, https://doi.org/10.5194/acp-21-10215-2021, 2021.
- Zobrist, B., Soonsin, V., Luo, B. P., Krieger, U. K., Marcolli, C., Peter, T., and Koop, T.: Ultra-slow water diffusion in aqueous sucrose glasses, Phys. Chem. Chem. Phys., 13, 3514–3526, https://doi.org/10.1039/c0cp01273d, 2011.

[4] There are several assumptions made in estimating the effect of particle viscosity on the N2O5 uptake in Sect. 2.5 that require further explanation and/or justification (e.g., choice of numeric values for various parameters in eqs. 3 and 4). As the effect on N2O5 uptake is prominently advertised in the manuscript title, this Section will need to be much improved.

[A4] We thank the reviewer for this constructive comment. In the revised manuscript, we have clarified the assumptions underlying Eqs. (S5.1) and (S5.2) within the uptake framework. In the main text (Sect. 2.6), we now briefly summarize the approach, while the full mathematical formulation and detailed derivations are provided in the Supplementary Information (Sect. S5). Importantly, this framework and the governing equations follow the same resistor-model formulation and parameterization previously used and evaluated in our earlier study (Song et al., 2025) and are not newly introduced in this work.

We have expanded the explanatory text in Sect. S5 to describe the physical meaning and origin of each parameter appearing in the uptake equations, including the accommodation coefficients, surface and bulk resistance terms, the viscosity-to-diffusivity conversion, and the characteristic particle length scales. For each parameter, we have indicated whether the chosen value is adopted from prior laboratory or modeling studies, represents a physically motivated bound, or follows directly from Stokes–Einstein or fractional Stokes–Einstein considerations.

In the revised manuscript, we have clarified that the $N_2O_5$ uptake calculation is intended as an illustrative application of the experimentally constrained viscosity parameterization rather than as a central quantitative result. To avoid overstating this component, Sect. S5 has been revised through structural revisions to the content and equations, along with refinements to the wording. Additional text has also been added in the revised manuscript (around lines 169–172).

"The $N_2O_5$ uptake calculations presented here are intended to illustrate how the experimentally constrained viscosity parameterization influences reactive uptake across tropospheric RH and

temperature conditions. These estimates should be interpreted in light of the limitations of this study, including the extrapolation of viscosity beyond the experimental temperature range (273–303 K).”

**Reference:**

- Song, M., Li, Y., Seong, C., Yang, H., Jang, K.-S., Wu, Z., Lee, J. Y., Matsuki, A., and Ahn, J.: Direct observation of liquid–liquid phase separation and core–shell morphology of PM2.5 collected from three northeast Asian cities and implications for N2O5 hydrolysis, ACS ES&T Air, 2, 1079–1088, https://doi.org/10.1021/acsestair.5c00043, 2025

[5] I am unclear about the choice of the content of Sect. 4 on the atmospheric implications. Why are only the estimates on the N2O5 uptake included here and not also the content of the current Sect. 3.2? Combining these two Sections (3.2 and 4.1) into one "Atmospheric implications" section and providing some further discussion on possible caveats on extrapolating their measurements to atmospheric conditions and particles would be helpful.

[A5] As suggested, we have combined the Sect. 3.2 and Sect. 4.1 into Sect. 3.2. In addition, we have explicitly incorporated a discussion of the key limitations of this study, which were raised by Reviewer 1. In particular, we have clarified the uncertainties associated with (i) extrapolating the viscosity parameterization beyond the experimental temperature range of 273–303 K, (ii) the strong sensitivity of viscosity to RH under cold tropospheric conditions, and (iii) the use of sucrose as a simplified proxy for chemically complex atmospheric organic aerosol. These caveats are now discussed directly within the revised section. Finally, we have included conclusion section in the revised manuscript.

**Specific comments:**

[6] L31-33: There are plenty of more recent (review) articles for this statement and much work has been done since the publication of the cited works. Please consider adding some more recent references to this statement, such as, but not limited to, e.g. (Zhang et al., 2021).

[A6] As suggested, the following references have been added to the revised manuscript (Line 32–34).

- Bei, N., Xiao, B., Wang, R., Yang, Y., Liu, L., Han, Y., and Li, G.: Impacts of aerosol–radiation and aerosol–cloud interactions on a short-term heavy-rainfall event–a case study in the Guanzhong Basin, China, Atmos. Chem. Phys., 25, 10931–10948, https://doi.org/10.5194/acp-25-10931-2025, 2025.

- El Haddad, I., Vienneau, D., Daellenbach, K. R., Modini, R., Slowik, J. G., Upadhyay, A., Vasilakos, P. N., Bell, D., de Hoogh, K., and Prevot, A. S. H.: Opinion: How will advances in aerosol science inform our understanding of the health impacts of outdoor particulate pollution?, Atmos. Chem. Phys., 24, 11981–12011, https://doi.org/10.5194/acp-24-11981-2024, 2024.

- Manavi, S. E. I., Aktypis, A., Siouti, E., Skyllakou, K., Myriokefalitakis, S., Kanakidou, M., and Pandis, S. N.: Atmospheric aerosol spatial variability: Impacts on air quality and climate change, One Earth, 8, https://doi.org/10.1016/j.oneear.2025.101237, 2025.

- McNeill, V. F.: Atmospheric aerosols: clouds, chemistry, and climate, Annu. Rev. Chem. Biomol. Eng., 8, 427–444, https://doi.org/10.1146/annurev-chembioeng-060816-101538, 2017.

- Nault, B. A., Jo, D. S., McDonald, B. C., Campuzano-Jost, P., Day, D. A., Hu, W. W., Schroder, J. C., Allan, J., Blake, D. R., Canagaratna, M. R., Coe, H., Coggon, M. M., DeCarlo, P. F., Diskin, G. S., Dunmore, R., Flocke, F., Fried, A., Gilman, J. B., Gkatzelis, G., Hamilton, J. F., Hanisco, T. F., Hayes, P. L., Henze, D. K., Hodzic, A., Hopkins, J., Hu, M., Huey, L. G., Jobson, B. T., Kuster, W. C., Lewis, A., Li, M., Liao, J., Nawaz, M. O., Pollack, I. B., Peischl, J., Rappenglück, B., Reeves, C. E., Richter, D., Roberts, J. M., Ryerson, T. B., Shao, M., Sommers, J. M., Walega, J., Warneke, C., Weibring, P., Wolfe, G. M., Young, D. E., Yuan, B., Zhang, Q., de Gouw, J. A., and Jimenez, J. L.: Secondary organic aerosols from anthropogenic volatile organic compounds contribute substantially to air pollution mortality, Atmos. Chem. Phys., 21, 11201–11224, https://doi.org/10.5194/acp-21-11201-2021, 2021.

- Su, H., Cheng, Y. F., and Pöschl, U.: New multiphase chemical processes influencing atmospheric aerosols, air quality, and climate in the Anthropocene, Acc. Chem. Res., 53, 2034–2043, https://doi.org/10.1021/acs.accounts.0c00246, 2020.

- Sun, Y. L., Luo, H., Li, Y., Zhou, W., Xu, W. Q., Fu, P. Q., and Zhao, D. F.: Atmospheric organic aerosols: online molecular characterization and environmental impacts, npj Clim. Atmos. Sci., 8, https://doi.org/10.1038/s41612-025-01199-2, 2025.
- Wall, C. J., Norris, J. R., Possner, A., Mccoy, D. T., Mccoy, I. L., and Lutsko, N. J.: Assessing effective radiative forcing from aerosol-cloud interactions over the global ocean, Proc. Natl. Acad. Sci. U.S.A., 119, https://doi.org/10.1073/pnas.2210481119, 2022.
- Zhang, Y., Liu, P., Han, Y., Li, Y., Chen, Q., Kuwata, M., and Martin, S. T.: Aerosols in Atmospheric Chemistry, ACS In Focus, American Chemical Society, https://doi.org/10.1021/acsinfocus.7e5020, 2021.

[7] L35: It could be meaningful to specify as "amorphous semi-solid" and "amorphous solid (or glassy)", at least here when you first introduce the different phase states, to avoid confusion with crystalline solid phase states that OA can also exist in.

[A7] As suggested, we have clarified that "OA can exist in liquid, semi-solid, and solid (amorphous or glassy) phase states" in the revised manuscript.

[8] L36: Consider introducing the symbol (η) used to represent dynamic viscosity already here and then use it throughout.

[A8] Because using the symbol η in the running text tended to make the notation inconsistent and less readable, we decided not to introduce η as a formal symbol in the main text. Instead, in the revised manuscript, we consistently use the words "viscosity" or "dynamic viscosity" in the running text and reserve symbols for the equations, ensuring unambiguous terminology.

[9] L38-39: "Accurate determination of aerosol viscosity and phase state…." Here and elsewhere (e.g., L50) it is unclear to me if you use "viscosity" and "phase state" synonymously or not. Please clarify.

[A9] Thank you for pointing this out. We do not use "viscosity" and "phase state" synonymously. In our framework, viscosity is treated as a quantitative material property, while particle phase state (e.g., liquid, semi-solid, glassy) is defined based on viscosity values. This definition was already adopted in the manuscript, and we have clarified the wording at its first occurrence to make this relationship explicit (Line 41).

[10] L45: Consider adding (Shiraiwa et al., 2017)

[A10]  The following reference has been added to the revised manuscript.

- Shiraiwa, M., Li, Y., Tsimpidi, A. P., Karydis, V. A., Berkemeier, T., Pandis, S. N., Lelieveld, J., Koop, T., and Pöschl, U.: Global distribution of particle phase state in atmospheric secondary organic aerosols, Nat. Commun., 8, 15002–15002, https://doi.org/10.1038/ncomms15002, 2017.

[11] L50: Please add (Koop et al., 2011)

[A11]  The following reference has been added to the revised manuscript.

- Koop, T., Bookhold, J., Shiraiwa, M., and Pöschl, U.: Glass transition and phase state of organic compounds: dependency on molecular properties and implications for secondary organic aerosols in the atmosphere, Phys. Chem. Chem. Phys., 13, 19238–19255, https://doi.org/10.1039/c1cp22617g, 2011.

[12] L54: Please add (Petters et al., 2019)

[A12]  The following reference has been added to the revised manuscript.

- Petters, S. S., Kreidenweis, S. M., Grieshop, A. P., Ziemann, P. J., and Petters, M. D.: Temperature- and humidity-dependent phase states of secondary organic aerosols, Geophys. Res. Lett., 46, 1005–1013, https://doi.org/10.1029/2018gl080563, 2019.

[13] L57-63: You might find the following helpful: (Li et al., 2020; Li and Knopf, 2021)

[A13]  The following references have been added to the revised manuscript.

- Li, J. N., Forrester, S. M., and Knopf, D. A.: Heterogeneous oxidation of amorphous organic aerosol surrogates by O3, NO, and OH at typical tropospheric temperatures, Atmos. Chem. Phys., 20, 6055–6080, https://doi.org/10.5194/acp-20-6055-2020, 2020a
- Li, J. N. and Knopf, D. A.: Representation of multiphase OH oxidation of amorphous organic aerosol for tropospheric conditions, Environ. Sci. Technol., 55, 7266–7275, https://doi.org/10.1021/acs.est.0c07668, 2021.

[14] L60: Please specify what you mean with "reactivity"

[A14] Thank you for this comment. By "reactivity," we specifically refer to the heterogeneous uptake and loss of $N_2O_5$ on particle surfaces, which is commonly quantified by the $N_2O_5$ uptake coefficient. We have revised the text accordingly for clarity. The revised sentence now reads (lines : 62–65)

"Modeling and observations have indicated that the presence of viscous organic shells or surfactant-rich surface layers can reduce the $N_2O_5$ uptake coefficient compared to liquid particles (McNeill et al., 2006; Cosman et al., 2008; Gaston et al., 2014; Ryder et al., 2015; Song et al., 2025)."

**References:**

- Cosman, L. M., Knopf, D. A., and Bertram, A. K.: N2O5 reactive uptake on aqueous sulfuric acid solutions coated with branched and straight-chain insoluble organic surfactants, J. Phys. Chem. A, 112, 2386–2396, https://doi.org/10.1021/jp710685r, 2008.

- Gaston, C. J., Thornton, J. A., and Ng, N. L.: Reactive uptake of N2O5 to internally mixed inorganic and organic particles: the role of organic carbon oxidation state and inferred organic phase separations, Atmos. Chem. Phys., 14, 5693–5707, https://doi.org/10.5194/acp-14-5693-2014, 2014.

- McNeill, V. F., Patterson, J., Wolfe, G. M., and Thornton, J. A.: The effect of varying levels of surfactant on the reactive uptake of N2O5 to aqueous aerosol, Atmos. Chem. Phys., 6, 1635–1644, https://doi.org/10.5194/acp-6-1635-2006, 2006

- Ryder, O. S., Campbell, N. R., Morris, H., Forestieri, S., Ruppel, M. J., Cappa, C., Tivanski, A., Prather, K., and Bertram, T. H.: Role of organic coatings in regulating N2O5 reactive uptake to sea spray aerosol, J. Phys. Chem. A, 119, 11683–11692, https://doi.org/10.1021/acs.jpca.5b08892, 2015

- Song, M., Li, Y., Seong, C., Yang, H., Jang, K.-S., Wu, Z., Lee, J. Y., Matsuki, A., and Ahn, J.: Direct observation of liquid–liquid phase separation and core–shell morphology of PM2.5 collected from three northeast Asian cities and implications for N2O5 hydrolysis, ACS ES&T Air, 2, 1079–1088, https://doi.org/10.1021/acsestair.5c00043, 2025.

[15] L66: I think the year for the Bateman study should be 2015, please check: https://pubs.acs.org/doi/10.1021/jp508521c.

[A15]   The reference has been corrected.

[16] L82: Renbaum-Wolff et al. 2013a and Renbaum-Wolff et al. 2013b refer to the same paper, please correct.

[A16] Thank you for the comment. Renbaum-Wolff et al. (2013a) and Renbaum-Wolff et al. (2013b) are two distinct publications, and the citations have been checked and are correctly listed in the revised manuscript.

[17] L89: What do you mean with "thermodynamic conditioning"? Please specify. (Also on L103)

[A17] Thank you for pointing this out. This comment addresses the same underlying concern raised by Reviewer 1 [1] regarding whether the droplets reached equilibrium with the surrounding gas phase prior to viscosity measurements.

To avoid ambiguity, we have removed the term "thermodynamic conditioning" throughout the manuscript and have replaced it with an explicit description of the experimental procedure. In the revised manuscript, we now state that droplets were held at the target relative humidity and temperature for sufficient time to allow equilibration with the surrounding gas phase and to ensure thermodynamic equilibrium prior to measurements. This clarification has been applied consistently at all relevant locations (including Lines 93–98).

[18] Fig. S1: Caption: Red line is not dotted, but solid, please fix. Please also add information on the goodness of fit to the figure, e.g., RMSE and R2.

[A18] The caption has been corrected, and information on the goodness of fit has been included.

 "Figure S1: Calibration line illustrating the relationship between mean bead speeds and viscosities of sucrose-$H_2O$ droplets at varying relative humidity (RH) levels. A linear regression, shown by the red solid line, fits the data with the equation: $viscosity = 0.00003 \times (mean\ bead\ speed)^{-0.971}$. The pink area denotes 95% prediction bands of fitting to the data in this study. The uncertainty in mean bead speed along *the x–axis* is calculated from standardization of 2–5 beads within 3–5 particles for each RH value."

[19] Fig. S2: The colored position labels appear very blurry and unreadable in most of the images. Please fix.

[A19] We have revised the figures, improved the resolution, and adjusted the font size and color of the position labels to ensure that they are now clear and easily readable. The updated figure is reproduced here:

[Figure]

Figure S2: Optical images of sucrose-$H_2O$ droplets during a typical bead-mobility experiment at different temperatures. Three labeled beads with tracked *x* and *y* coordinates were used to determine average bead speeds using ImageJ software. The size of the scale bar is 20 μm.

[20] Fig. S3a/b: - Caption: "… the uncertainty in RH from the RH sensor during calibration at each temperature…" I do not follow this. Why is the uncertainty in RH not the same across all data points? Is this uncertainty not given by the error/uncertainty in your RH-sensor? Please specify in the text how the RH uncertainty was determined more clearly by giving a reference to Section S1, where your deliquescence values are described.

[A20] Thank you for the comment. We have corrected the caption of Figure S3 as follows:

"Figure S3: (a) Mean bead speed and (b) resulting viscosities from bead-mobility experiments for sucrose-$H_2O$ droplets as a function of temperature and relative humidity (RH). The $x$-axis error bars represent the RH range in a given experiment and the uncertainty in RH measurements. The $y$-axis error bars indicate the standard deviation of the measured bead speeds and viscosity calculated from 3–5 beads across 2–5 particles at each RH level."

Why is there only a single data point for 273 K?

The reason there is just one data point at 273 K is that the beads movement became so slow that it could not be measured accurately, making it impossible to determine their speeds at RH levels < ~75% and resulting in only one data point at this temperature.

[21] L110: Fig. S4: I cannot really see an evolution of the particle morphology in these panels. For each RH/T combination only 1 image before and 1 image after poking is shown. It could help to add one more column of images, where the particle is shown after a fixed time after poking (e.g. 10 s) for ALL RH/T combinations. Right now, each RH/T combination shows a different experimental flow time, which makes a comparison very hard.

[A21] For clarity, we have added lines to better distinguish the evolution of the hole morphology for Fig. S4.

[Figure]

| Temp | Pre-poking | Poking | Post-poking | $\tau_{exp,flow}$ |
|---|---|---|---|---|
| 273K, ~ 50% RH | | | 0 sec | 6768 sec |
| 283 K, ~ 50% RH | | | 0 sec | 397 sec |
| 293 K, ~ 50% RH | | | 0 sec | 25 sec |
| 303 K, ~ 50% RH | | | 0 sec | 3 sec |

Figure S4: Optical images for experimental flow times ($\tau_{exp,flow}$) during poke-and-flow experiments at different temperatures. The white scale bar indicates 20 μm. The red dotted lines indicate the size of the cavity measured at the corresponding time after poking.

[22] Fig. S5: Caption. Add "." After Sellier et al. (2015).

[A22] The "." has been added in the revised manuscript.

[23] L119: Specify: "to temperature changes"

[A23] The text has been revised as follows (lines 128–130):

"To determine the fragility parameter ($D_f$), which describes how sensitively viscosity responds to temperature changes, we fitted our experimentally measured viscosity data, spanning a temperature range of 273–303 K and RH ~20–90%, to the VFT equation (Eq. 1)."

[24] L121: Here and in some other equations the formatting of "ln" is off, as it is italicized in some eqs. and not in others. Cf. dominator vs. denominator in eq. (2). Please fix.

[A24] The equation has been updated with the correct use of the symbols.

[25] L123: What is "infinite temperature"? Please also move the reference to Angell 1991 in front of the comma.

[A25] As addressed in our response [A11] to Reviewer 1, $\eta_\infty$ in Eqs. (1)–(2) is the VFT pre-exponential factor (high-temperature limiting viscosity as $T \to \infty$), not a viscosity measured at an "infinite temperature." We revised the manuscript accordingly and moved the Angell (1991) citation to precede the comma, as suggested.

[26] L125: Why have a "(293K)" in the nominator?

[A26] The appearance of 293 K reflects our choice of a reference temperature at which the relationship between viscosity and composition is evaluated. In our approach, the RH-dependent Vogel temperature $T_0$(RH) is first determined using viscosity measurements at 293 K, after which the full VFT formulation is applied to predict viscosities at other temperatures. This procedure follows the same methodology adopted in our previous study (Song et al., 2025), where 293 K was likewise used as a reference temperature to anchor the composition dependence before extending the VFT model across temperature. We have clarified this point in the revised manuscript for consistency and transparency.

**Reference:**

- Song, M., Li, Y., Seong, C., Yang, H., Jang, K.-S., Wu, Z., Lee, J. Y., Matsuki, A., and Ahn, J.: Direct observation of liquid–liquid phase separation and core–shell morphology of PM2.5 collected from three northeast Asian cities and implications for N2O5 hydrolysis, ACS ES&T Air, 2, 1079–1088, https://doi.org/10.1021/acsestair.5c00043, 2025

[27] L129-133: It is unclear how you determined a fragility parameter from your Fig. S8. A detailed description of the steps how the experimental data was used to determine the fragility parameter should be added to the SI of this work. Looking over the work by Derieux et al. (2018) cited by the authors, it appears to me that they suggest a "lower limit fragility parameter of 10 ± 1.7". This does not seem too far off from the reported value of 13 ± 1. A more detailed description of what the fragility parameter means for the interpretation of your data and those from previous work should be added to better rationalize differences between the fragility parameter reported by you and others.

[A27] See our response to General Comment [2]. We added a detailed description of the global fitting procedure used to derive $D_f$ (Fig. S8) and expanded the comparison with literature $D_f$ values in Sect. 2.4 and the SI.

[28] L134: Please replace "Simulation" with "Estimation of" here and elsewhere.
[A28] We have revised as suggested.

[29] L135: Here and elsewhere, please check placement of references and punctuation.
[A29] The punctuation and reference placement have been checked and corrected throughout the revised manuscript.

[30] Sect. 2.5: The various assumptions made in this section should be explained in more detail. It is unclear to me how the chosen parameters fit to your experimental conditions. For example, the authors use a coefficient for the surface reaction of 2.5 x 10-4 bas on the "low RH conditions" by Grzinic et al. 2015. Are these low RH conditions representative for the entire RH-range probed by you? Similar, an explanation should be given, why αb = 0.035 can be chosen.
[A30] See Response [A4] above for the revised text in Sect. 2.6, where the approach is summarized in the main text, with detailed justification of parameter choices and their applicability across the relative humidity range provided in Sect. S5.

[31] L154: "The result shows…" This statement is misleading. You might observe viscosity changes over 9 orders of magnitude when comparing across the RH range from ~10-90%, but for any given RH value, the range of viscosities observed across the different temperatures tested is much less than 9 orders or magnitude. Please reformulate more carefully.
[A31] Thank you for pointing this out. The updated text in the revised manuscript (around lines 177–178) is given as:
"The results show that, across this combined temperature–RH space, the viscosity of sucrose-$H_2O$ droplets spans approximately eight orders of magnitude."

[32] Fig. 1: Caption: It is unclear to me, how the colored, dashed fit-lines were established. Are these fits constrained to the viscosity of water at 100 % RH? Is this a fit to just the data points of this study or are literature values taken into account where available, i.e., at 293 K?

[A32] The colored dashed lines in Figure 1 are a simple linear least-squares fit to the viscosity data obtained in this study at each temperature. The fits are not constrained to pass through the viscosity of liquid water at 100% RH, and we did not include any external literature data (for example, values at 293 K) when determining these lines. To make it clearer, we revised the caption as: "Dotted lines represent linear fits to the viscosity data obtained in this study at each temperature."

[33] L164: "…, indicating a semi-solid state." I think the reasoning or cause-effect relationship is off here. The fact that the reduction in viscosity at 50% RH was lager when going from 273 K 303 K as compared to at 80% RH does not allow you to make statements about the phase state. I.e., it is not the change in viscosity but the absolute viscosity values that allow you to make statement about the aerosol phase state. Please improve the description. Maybe I am misinterpreting what you want to say here?

[A33] Thank you for pointing this out. We have rephrased the sentence as follows (Lines 186–188):

"At mid RH (~50%), increasing the temperature from 273 K to 303 K reduced the viscosity from ~2.1 × 10$^6$ Pa·s at 273 K to ~1.4 × 10$^3$ Pa·s at 303 K, remaining within the semi-solid regime over the investigated temperature range."

[34] L165: Please see my comment above. "Flow time contrast" for what? To fully recover to the initial particle morphology after poking? Please clarify.

[A34] To avoid confusion, we have removed this sentence from the revised manuscript.

[35] L170: Should the viscosity threshold for a "glassy" phase state not be > 10$^{12}$ Pa s?

[A35] We have revised the manuscript (around lines 193–195) to clarify this distinction as:

"Once a fracture occurred, no restorative flow was observed for more than two hours, indicating the lower-limit viscosity of ~1 × 10$^8$ Pa·s, consistent with the particle behaving as a semisolid or solid (Renbaum-Wolff et al., 2013b; Grayson et al., 2015; Song et al., 2019)."

**References:**

- Renbaum-Wolff, L., Grayson, J. W., Bateman, A. P., Kuwata, M., Sellier, M., Murray, B. J., Shilling, J. E., Martin, S. T., and Bertram, A. K.: Viscosity of α-pinene secondary organic material and implications for particle growth and reactivity, Proc. Natl. Acad. Sci. U.S.A., 110, 8014–8019, https://doi.org/10.1073/pnas.1219548110, 2013b.

- Grayson, J. W., Song, M., Sellier, M., and Bertram, A. K.: Validation of the poke-flow technique combined with simulations of fluid flow for determining viscosities in samples with small volumes and high viscosities, Atmos. Meas. Tech., 8, 2463–2472, https://doi.org/10.5194/amt-8-2463-2015, 2015.

- Song, M., Maclean, A. M., Huang, Y., Smith, N. R., Blair, S. L., Laskin, J., Laskin, A., DeRieux, W. S. W., Li, Y., Shiraiwa, M., Nizkorodov, S. A., and Bertram, A. K.: Liquid–liquid phase separation and viscosity within secondary organic aerosol generated from diesel fuel vapors, Atmos. Chem. Phys., 19, 12515–12529, https://doi.org/10.5194/acp-19-12515-2019, 2019.

[36] L185: "Figure 2…" This sentence is incomplete, please fix.

[A36] As suggested, we have revised the incomplete sentence. The text now reads:

"Figure 2 illustrates the resulting RH–temperature phase diagram, in which contour lines (isopleths) represent levels of constant viscosity."

[37] L201: "Viscosity isopleths…" This sentence is incomplete, please fix.

[A37] As suggested, we have revised the incomplete sentence. The text now reads:

"Viscosity isopleths were computed using the VFT parametrization, with the $D_f$ obtained by fitting the experimental viscosity data."

[38] L183: It could be helpful to add a reference to your Section S2 and/or your eq. (S2.1) here, to clarify how the RH and T-dependent viscosity values were derived.

[A38] As suggested, the text has been revised as follows:

"We then extended the results with the VFT equation (Eqs. 1 and 2) and the viscosity parameterization described in Sect. S2 (see Eq. S2.1) to calculate viscosity across a wider range…."

[39] L188: "…the contours are nearly vertical". While true, it could be more appropriate to talk about a slope here, as you do further down. To be more quantitative, you could consider discussing the slope of the black, dashed line (semi-solid/liquid transition) across the 3 temperature ranges that you discuss between L187-193.

[A39] Thank you for the comment. As suggested, the text has been updated in terms of the slope, and the revised text (lines 212–220) is provided below:

"The orientation of these isopleths varied systematically with temperature. At low temperatures ($T$ < 240 K, representative of the upper troposphere, ~8–12 km), the isopleths exhibited very steep slopes in RH–temperature space, indicating that viscosity was dominated by RH such that even small increases in RH lead to orders-of-magnitude decreases in viscosity. In this regime, the liquid–semi-solid phase boundary (black dashed line) likewise showed a steep slope, reflecting the strong RH sensitivity of the phase transition. Between 240 and 280 K (typical of the middle troposphere, ~3–8 km), the isopleths displayed more moderate slopes, indicating a combined influence of both temperature and RH on viscosity and phase state. At higher temperatures (280–310 K, representative of the lower troposphere, ~0–3 km), the isopleths and the phase boundary showed gentler slopes, consistent with reduced RH sensitivity and a comparatively stronger temperature control on viscosity."

[40] L207: Add "… we applied the viscosity parameterization (eq. S2.1)…"
[A40] As suggested, we have added the equation number.

[41] L207-208: Do you refer to the RH and T fields shown in Fig. S9? If yes, a reference would be appropriate. The authors should also add further information about where this RH and T-information was retrieved from. It is stated that this is re-analysis data. Re-analysis of what data? The authors should also clarify why only a period of 2020-2024 was used to determine the average RH and T fields and why they consider this representative?

[A41] Yes. As addressed in our response to Reviewer 2 [1] (and Reviewer 1 [3]), the RH and temperature fields shown in Fig. S9 are monthly mean ECMWF ERA5 pressure-level reanalysis data (Copernicus Climate Data Store) averaged over 2020–2024, zonally averaged, and converted to geometric altitude using Eq. S4.1 (details in Sects. 2.5 and S4).

[42] L209-201: "… derived from monthly mean ambient RH and T …" Could this not be simplified here in the main text by saying that these are average RH and T profiles over the time period 2020 to 2024? It also seems inconsistent to write about monthly averages here and annual averages on L240.

[A42] We have revised the sentence as follows:

"……employing multi-year monthly mean fields averaged from 2020 to 2024 to represent recent tropospheric conditions…."

[43] L211: Add "fractional Stokes-Einstein…"

[A43] The fractional Stokes-Einstein equation has been added.

[44] L218: Please quantify "extremely high RH".

[A44] We have quantified in the revised text by specifying the RH range as > 80% in these regions. The text now reads:

"This persistence was consistent with the high RH (> 80%) observed in these regions…."

[45] L219: Viscosity is not "suppressed", rather does the presence of water vapor and water in the particles prevent the formation of highly viscous phase states. Please improve formulation.

[A45] The sentence has been revised accordingly (lines 244–246).

"This persistence was consistent with the high RH (> 80%) observed in these regions (Fig. S9), which enhanced water uptake and plasticization of the organic matrix, thereby delaying the formation of highly viscous or glassy phase states."

[46] L225-226: "… together these factors tend …" I am not sure I follow this sentence. What are multi-component SOA fields?

[A46] To improve clarity and avoid overinterpretation, we have removed this sentence from the revised manuscript.

[47] L230: Can you give some examples of these chemical transport models and provide some references for the assumption of fast mixing?

[A47] The sentences in the revised manuscript have been updated as follows (lines 258–263):

"This contrasts with the approach adopted in regional and global atmospheric models (e.g., EMAC, GEOS-Chem), which generally assume that semi-volatile organic compounds mix and equilibrate within particles on timescales comparable to the model time step (typically sub-hour to ~1 h) (Maclean et al., 2017; Shiraiwa et al., 2017; Luu et al., 2025). Under conditions where our calculated mixing times substantially exceed these timescales, such models are therefore likely to underestimate kinetic limitations in the troposphere."

References:

- Maclean, A. M., Butenhoff, C. L., Grayson, J. W., Barsanti, K., Jimenez, J. L., and Bertram, A. K.: Mixing times of organic molecules within secondary organic aerosol particles: a global planetary boundary layer perspective, Atmos. Chem. Phys., 17, 13037–13048, https://doi.org/10.5194/acp-17-13037-2017, 2017.

- Luu, R., Schervish, M., June, N. A., O'Donnell, S. E., Jathar, S. H., Pierce, J. R., and Shiraiwa, M.: Global simulations of phase state and equilibration time scales of secondary organic aerosols with GEOS-Chem, ACS Earth Space Chem., 9, 288–302, https://doi.org/10.1021/acsearthspacechem.4c00281, 2025

- Shiraiwa, M., Li, Y., Tsimpidi, A. P., Karydis, V. A., Berkemeier, T., Pandis, S. N., Lelieveld, J., Koop, T., and Pöschl, U.: Global distribution of particle phase state in atmospheric secondary organic aerosols, Nat. Commun., 8, 15002–15002, https://doi.org/10.1038/ncomms15002, 2017.

[48] L233: Add "as a proxy for (secondary?) organic aerosol. However, real…"

[A48] Yes, we have added it in the revised manuscript.

[49] L236: I would probably delete "under humid conditions". As most atmospheric inorganic substances undergo deliquescence and efflorescence, it will dependent on the history of the particle, whether the inorganic components of an internally mixed organic-inorganic aerosol are dissolved and hence can promote water uptake or not.

[A49] The phrase "under humid conditions" has been removed, and the sentence has been revised accordingly.

[50] Sect. 4.1: General: It is not entirely clear to me how you use the RH- and T-dependent viscosity data to calculate the N2O5 uptake. Going back to L148-149 it seems like you use the viscosities along with the fractional-Stokes Einstein relationship to calculate N2O5 diffusion coefficients and then use these calculated diffusion coefficients in eqs. (4) and (3), as described in SI L53-5. It would be helpful to make this clearer in the main text, e.g., on L148-149 and maybe also reiterate at the beginning of Sect. 4.1.

[A50] As clarified in revised Sect. 2.6 (see also Response [A4] here), we now explicitly state the workflow $\eta(RH,T) \rightarrow D_{N_2O_5}$ (fractional Stokes–Einstein) $\rightarrow \gamma_{N_2O_5}$ in the main text, with the corresponding equations provided in the Supplementary Information (Eqs. S5.1–S5.2).

[51] L251: "… could extend over several shorts of the year…" It is unclear what you mean here, please rephrase.

[A51] As also addressed in our response to Reviewer 1 [23], the phrase "several shorts" was an unintended typo and has been rephrased in the revised manuscript as follows (lines 269–270): "Using our parameterized viscosity fields, we estimated that in the midlatitude upper troposphere, suppressed uptake could extend over several short periods, particularly in winter."

[52] L252: Please quantify "cold conditions"

[A52] "Cold conditions" have been quantified in the revised text. The sentence now reads: "This aligns with recent field observations showing reduced $N_2O_5$ loss rates in cold conditions ($\leq$ ~0 °C)."

[53] L257: "uptake rates of" delete "of"

[A53] The "of" has been deleted.

[54] L260: "As shown in Fig. 3c" This sentence seems incomplete and is hard to understand, please reformulate.

[A54] We have revised the sentence as: "As shown in Fig. 3c, $N_2O_5$ uptake coefficient is leveled off near ~$10^{-3.5}$ at higher altitudes, because the surface reaction term ($\Gamma_s$) was fixed at $2.5 \times 10^{-4}$ in Eq. (S5.1)."

[55] L263: For better comparison of your current Fig. 3c and Fig. S11, it would be helpful to have them next to each other. Consider splitting Fig. S11 into two panels, where one panel reproduces your Fig. 3c and the other panel displays the results of your sensitivity analysis.

[A55] Thank you for this helpful suggestion. We agree that side-by-side comparison can in principle aid visual interpretation. However, we have chosen to retain the current figure organization to avoid reproducing the same results in multiple figures.

Figure 3c already presents the reference $N_2O_5$ uptake results discussed in the main text, while Fig. S11 is intended to complement this figure by showing the outcomes of the sensitivity analysis. To maintain clarity and avoid redundancy, we therefore do not reproduce Fig. 3c within Fig. S11, but instead refer readers directly to Fig. 3c for comparison.

[56] L266: I would not call this a climatology, which usually would contain data over several decades. Rather use the wording of "average RH- and T-fields" or similar.

[A56] The wording of the sentence has been changed as suggested by the reviewer.

[57] L266: I would encourage the authors to tune down the wording here a little bit. There have been previous attempts to explore the phase state of organic aerosol throughout the troposphere, as evident by the studies cited in this work. Related, with this statement the authors seem to claim that the sucrose viscosity is representative for the viscosity of complex atmospheric aerosol, directly contradicting their statement on L236-237.

[A57] As suggested, we have toned down the wording to better acknowledge previous studies and avoid overstating the novelty of the present work (lines 283–285).

"By directly constraining temperature-dependent viscosity across a 30 K range and coupling it to global RH–and temperature fields, this study could provide novel experimental constraints for assessing OA phase behavior under tropospheric conditions."

**Supporting Information:**

[58] L32: Delete "." In front of "(Weight, 2019)"

[A58] The extra period has been deleted in front of (Weight, 2019).

[59] L36: Move "(Price et al. 2016)" in front of "(Price et al. 2016)"

[A59] The reference placement has been corrected as suggested.

[60] L58: Add "." After Kiland et al. (2023).

[A60] "." has been added after Kiland et al. (2023).

[61] L66: eq. (S2.3) what values of k was used here? It would be helpful to replace "k" by the Greek symbol "kappa" ($\kappa\kappa$) here and elsewhere, as traditionally used to describe aerosol hygroscopicity.

[A61] In Eq. (S2.3), the hygroscopicity parameter $\kappa$(formerly denoted as $k$) was used with a value of $\kappa = 0.061$.

It would be helpful to replace "k" by the Greek symbol "kappa" ($\kappa\kappa$) here and elsewhere, as traditionally used to describe aerosol hygroscopicity

As suggested, the hygroscopicity parameter is now denoted by the Greek symbol ($\kappa$) instead of k throughout the manuscript and Supplementary Information, following standard aerosol hygroscopicity notation.

[62] L75: "Petters" is misspelled.

[A62] Petters spelling has been corrected.

[63] L79: Numbering of eq. is off and should read "(S3.1)"

[A63] The equation numbering has been revised to be consistent with the updated section numbering.

[64] Fig. S7: It is unclear to me how you use eq. (S2.1) to determine a hygroscopicity value from this data. Please specify in the text.

[A64] This point is addressed in our response to Reviewer 1 [A12].

[65] Fig. S9: Please increase the size of this figure over the full page to allow the reader to read out values from the figure. In its current form it is extremely hard to read the figure. The authors could also consider removing the panels of the different seasons, if a season-dependent viscosity is not discussed.

[A65] The size of the figure has been increased to the full page.

[66] Fig. S10: Please increase the size of this figure over the full page to allow the reader to read out values from the figure. In its current form it is extremely hard to read the figure.

[A66] The size of the figure has been increased to the full page.

[67] L268: Why do the authors only mention the need for T-dependent viscosity parameterizations? What about the role of RH for organic aerosol viscosity, which according to your Fig. 2 is the key parameter in controlling the aerosol phase state in the troposphere.

[A67] We have revised the sentence as:

"…..temperature- and RH-sensitive viscosity parameterization……